# Chemical modulation of cytosolic BAX homodimer potentiates BAX activation and apoptosis

Nadege Gitego[1,2,3], Bogos Agianian[1,2,3], Oi Wei Mak[1,2,3], Vasantha Kumar MV[1,2,3], Emily H. Cheng [4,5,6] & Evripidis Gavathiotis [1,2,3] ✉

The BCL-2 family protein BAX is a major regulator of physiological and pathological cell death. BAX predominantly resides in the cytosol in a quiescent state and upon stress, it undergoes conformational activation and mitochondrial translocation leading to mitochondrial outer membrane permeabilization, a critical event in apoptosis execution. Previous studies reported two inactive conformations of cytosolic BAX, a monomer and a dimer, however, it remains unclear how they regulate BAX. Here we show that, surprisingly, cancer cell lines express cytosolic inactive BAX dimers and/or monomers. Expression of inactive dimers, results in reduced BAX activation, translocation and apoptosis upon pro-apoptotic drug treatments. Using the inactive BAX dimer structure and a pharmacophore-based drug screen, we identify a small-molecule modulator, BDM19 that binds and activates cytosolic BAX dimers and prompts cells to apoptosis either alone or in combination with BCL-2/BCL-XL inhibitor Navitoclax. Our findings underscore the role of the cytosolic inactive BAX dimer in resistance to apoptosis and demonstrate a strategy to potentiate BAX-mediated apoptosis.

The intrinsic or mitochondrial pathway of apoptosis is largely regulated by the BCL-2 family of proteins and their interactions[1,2]. The BCL-2 family includes the pro-apoptotic or effector proteins BAX, BAK, and BOK, the anti-apoptotic or survival proteins e.g., BCL-2, BCL-XL, MCL-1, and the pro-apoptotic BH3-only proteins categorized either as activators e.g., BIM, BID or sensitizers e.g., BAD, HRK[1,2]. Effector proteins have a key role in initiating apoptosis by their ability to undergo conformational transformation, when they are typically activated by activator BH3-only proteins, to induce mitochondrial outer membrane permeabilization (MOMP)[3–5]; a critical event in apoptosis that releases apoptogenic factors such as cytochrome c into the cytosol to promote the caspase cascade signaling of apoptosis[6,7]. Anti-apoptotic BCL-2 proteins can inhibit both effectors and activator BH3-only proteins to ensure cell survival[8]. Sensitizer BH3-only proteins interact with

anti-apoptotic BCL-2 proteins and release pro-apoptotic proteins that are blocked by anti-apoptotic BCL-2 proteins[9,10]. This enables the released pro-apoptotic BCL-2 proteins to proceed with the execution of MOMP and apoptosis[9,10].

Cancer cells typically ensure their survival, growth and resistance to various cancer treatments by upregulating anti-apoptotic BCL-2 proteins to overwhelm pro-apoptotic proteins[11,12]. This provides the means for cancer cells to block the activation of pro-apoptotic effectors BAX/BAK and apoptosis induction. Pro-apoptotic BAX is a crucial mediator of apoptosis induced by diverse stimuli including various chemotherapeutic and targeted agents[13,14]. BAX is predominantly localized in the cytosol in a quiescent conformation[15,16]. Inactive soluble BAX is a globular protein with nine α helices linked with variable loops[14,17]. BAX activation ensues upon binding of a BH3-only protein

[1]Department of Biochemistry, Albert Einstein College of Medicine, Bronx, NY, USA. [2]Department of Medicine, Albert Einstein College of Medicine, Bronx, NY, USA. [3]Montefiore Einstein Comprehensive Cancer Center, Albert Einstein College of Medicine, Bronx, NY, USA. [4]Human Oncology and Pathogenesis Program, Memorial Sloan Kettering Cancer Center, New York, NY, USA. [5]Department of Pathology and Laboratory Medicine, Memorial Sloan Kettering Cancer Center, New York, NY, USA. [6]Weill Cornell Medicine, New York, NY, USA. ✉e-mail: evripidis.gavathiotis@einsteinmed.edu

with its BH3 domain helix to the N-terminal BAX trigger site (α1, α6 helices), inducing conformational changes: the opening of the α1-α2 loop and exposure of the 6A7 antibody epitope, the mobilization of the BH3 domain (α2) and the C-terminal α9 helix from the hydrophobic core of BAX[18–22]. The mobilized α9 helix enables translocation and anchorage of BAX in the mitochondrial outer membrane, where BAX oligomerizes and induces MOMP[10,19–23]. Previous reports showed that BAX is regulated through its localization, binding partners, and post-translational modifications (PTMs)[24–29]. Specifically, BAX can shuttle between the cytosol and the mitochondria by interacting with anti-apoptotic BCL-2 proteins that reside at the outer mitochondrial membrane[30,31]. When BAX translocates to the mitochondria to induce MOMP, it can be sequestered by anti-apoptotic BCL-2 proteins forming stable heterodimers[9,12]. In addition, post-translational modifications such as phosphorylation or ubiquitination at key BAX residues can modulate its activity or promote its degradation, respectively[28,29]. However, post-translational modifications of BAX appear limited to specific cellular contexts.

While cytosolic BAX is typically considered an inactive monomer, we previously reported structural and cell-based evidence of cytosolic BAX dimer in an autoinhibited conformation[15–17,32]. This inactive BAX dimer is formed by the interaction of one BAX protomer interacting with the N-terminal trigger site and the second BAX protomer interacting with the C-terminal surface which mainly includes α9 helix[15]. This BAX dimer conformation hinders the N-terminal trigger site and the mobilization of α9 helix from the BAX core, which renders BAX less susceptible to conformational activation and thereof translocation and oligomerization. We previously found that specific BAX mutants such as P168G and G67R formed inactive BAX dimer in crystal structures and in vitro[15]. The P168G mutant was expressed as cytosolic BAX dimer in MEFs with impaired capacity to induce apoptosis. Particularly, mutations of P168A and G67R have been identified in AML patients and found to have impaired apoptotic activity and notably the BAX P168A mutant appeared in AML patients acquiring resistance to the BCL-2 inhibitor Venetoclax[33,34].

In this work, we investigate the role of the inactive BAX dimer in regulating apoptosis and pro-apoptotic drug treatments in various cancer cells, since understanding how BCL-2 family proteins are regulated in cancer and how they respond to various cancer treatments have important fundamental and translational implications. Our findings support the notion that the formation of the inactive BAX dimer is a mechanism adopted by cancer cells to further suppress BAX activation and gain a survival advantage. Using a pharmacophore-based screen, we identify a chemical probe to modulate the cytosolic inactive BAX dimer. We find BAX dimer modulator 19 (BDM19) that binds cytosolic BAX, activates cytosolic BAX dimers and prompts cells to apoptosis alone or in combination with BCL-2/BCL-XL inhibitor Navitoclax (ABT-263). Thus, our findings provide mechanistic insights into the regulation of BAX and resistance to apoptosis in cancer through inactive BAX dimerization. Moreover, we demonstrate a rational strategy to identify BAX dimer modulators and potentially a new class of pro-apoptotic drugs for cancer therapy.

## Results

### Diverse cancer cell lines express cytosolic inactive BAX dimer and BAX monomer

We previously established a cell fractionation protocol following cell lysis to isolate the cytosolic fraction from the mitochondrial fraction (see methods, Fig. 1a)[15]. Since it is established that BAX can be activated by non-ionic detergents, we used a buffer that contains no detergent aiming to avoid potential BAX activation that may lead to mitochondrial translocation and oligomerization. Using this protocol in MEFs, an established model cell line for apoptosis, and BAX mutations suggested by the crystal structure of the inactive BAX dimer conformation, we previously

demonstrated that cytosolic BAX may adopt an inactive BAX dimer or BAX monomer conformation[15,35]. To broadly investigate the role of the inactive BAX dimer in cancer cells, we performed fractionation of cytosolic fractions of a diverse panel of hematological and solid tumor cell lines, including leukemia, lymphoma, colorectal and non-small lung cancer cell lines, with various genomic alterations. Cytosolic fractions from different cell lines were analyzed by size-exclusion chromatography (SEC) followed by western blot for BAX detection to identify the size of cytosolic BAX (Fig. 1a, b). OCI-AML3 and HPB-ALL cells were found to express BAX monomer only (centered at 15.5–16 ml), but surprisingly, other cell lines express BAX dimer (centered at 14–14.5 ml) (Fig. 1b) and cell lines like U937 and SU-DHL5 may express both BAX dimer and monomer. Interestingly, non-cancerous cell lines BEAS-2B and IMR90 express only BAX dimer (Fig. 1b). In CALU-6, HCT-116, and Namalwa cells that express cytosolic BAX dimers, we did not detect any of the common anti-apoptotic BCL-2 proteins (e.g. BCL-XL, BCL-2 and MCL-1) in the cytosolic fractions but rather only in the mitochondrial fractions, suggesting that the cytosolic BAX dimers cannot correspond to heterodimers with anti-apoptotic BCL-2 proteins (Fig. 1c, Supplementary Fig. 1a). Moreover, BCL-XL was detected in the cytosolic fraction of SUDHL-16 cells, but SEC analysis showed that BAX and BCL-XL elute at separate fractions (Supplementary Fig. 1a, b). Interestingly, we quantified the total levels of BAX, and BAX dimers in cytosolic fractions of the analyzed cell lines (Fig. 1a), and we found a positive correlation of BAX dimers with total BAX, suggesting that protein levels of BAX may determine the cytosolic BAX conformation (Supplementary Fig. 1c).

We then established a Blue-native PAGE (BN-PAGE) assay to evaluate isolated cytosolic fractions for BAX conformations. Both CALU-6 and HCT-116 cells that express BAX dimers (Fig. 1b), showed cytosolic BAX that appeared predominantly at ~120 kDa in BN-PAGE (Fig. 1d). Previous studies showed that BAX appears at ~60 kDa when lysates containing inactive BAX monomer were separated by BN-PAGE[21,36]. The inactive BAX dimer has been characterized to be reversible into inactive BAX monomers and thereafter active BAX monomers and oligomers, by treatment with BIM SAHB, a stapled BH3 peptide that activates BAX through the N-terminal trigger site[15,18]. Interestingly, treatment of the cytosolic fraction of both CALU-6 and HCT-116 cells with BIM SAHB resulted in BAX bands at ~20 kDa and ~600 kDa in BN-PAGE (Fig. 1d). Consistent with previous analysis by BN-PAGE, active BAX monomer induced by a BIM BH3 peptide runs lower than the inactive BAX monomer[21,37]. BAX immunoprecipitation of the cytosolic fractions of both CALU-6 and HCT-116 cells by 6A7, an antibody that recognizes active BAX through an epitope that is otherwise occluded in inactive BAX, was negative[19,32]. However, treatment of the same cytosolic fractions with 1% triton-X, previously characterized to activate BAX, allowed successful BAX immunoprecipitation by 6A7, confirming that the observed cytosolic BAX dimer is indeed in an inactive conformation (Fig. 1e)[15]. Overall, these data suggest that in cells, cytosolic BAX can adopt an inactive dimer conformation that can be reduced and activated upon BIM BH3 triggering.

To determine the potential contribution of BAX conformation to apoptosis among cell lines that express cytosolic BAX dimers and monomers, we evaluated mitochondrial depolarization of cancer cell lines upon BIM BH3 peptide treatment and found cells with BAX dimers, in contrast to cells with BAX monomers, to be unprimed to apoptosis. (Supplementary Fig. 2a). Because cancer cell lines have different mutational landscape and expression levels of the BCL-2 family proteins, to evaluate the contribution of BAX dimer and monomer more precisely to apoptosis, we generated cells expressing cytosolic BAX dimer or monomer in the same cellular background. HCT116 BAX KO cells were transduced with human BAX WT, BAX

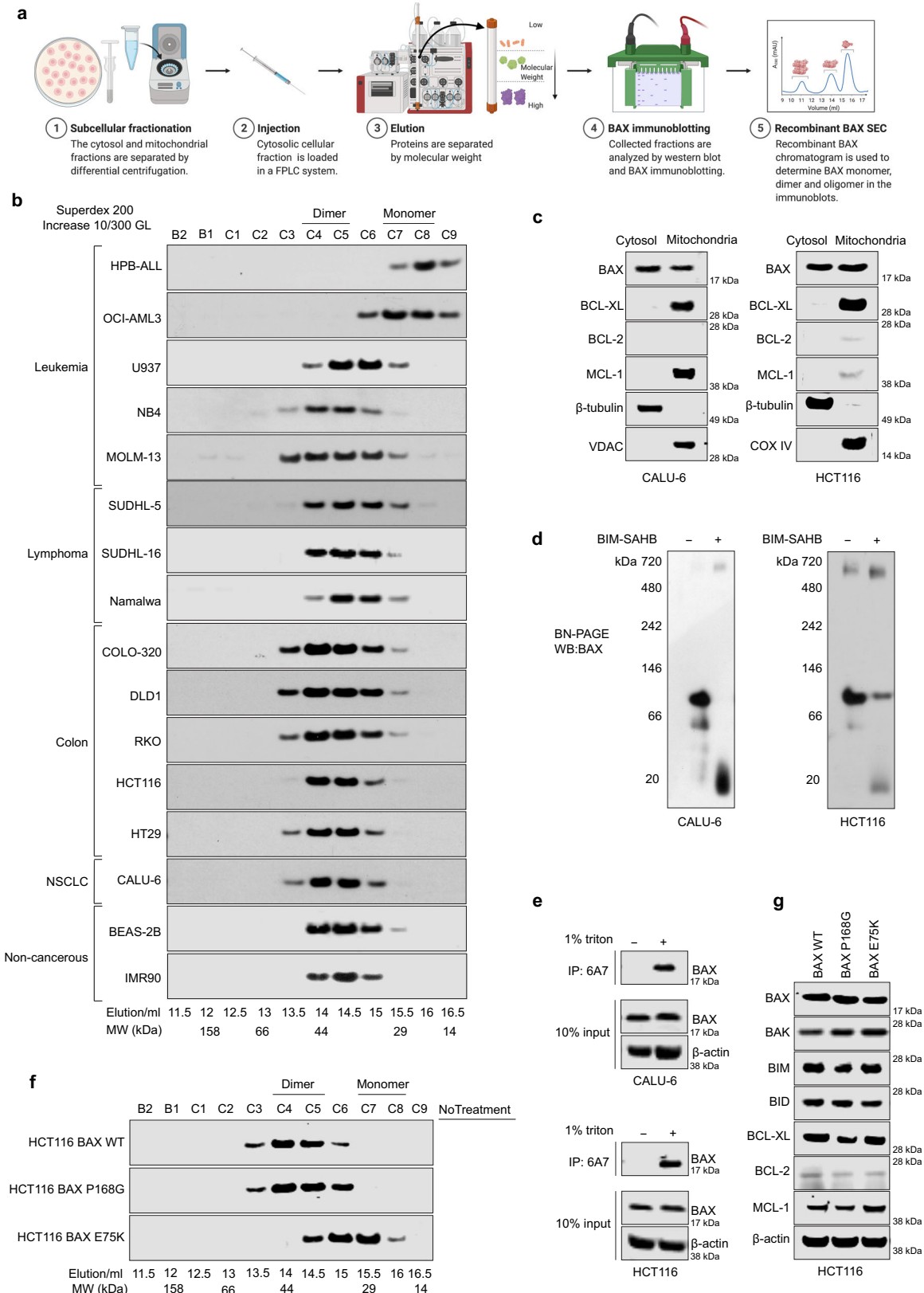

P168G (a dimerization stabilizing mutation) and BAX E75K (a BAX dimerization impairing mutation)[15] (Supplementary Fig. 2b). As predicted from the SEC profile of parental HCT116, HCT116 BAX KO reconstituted with BAX WT or BAX P168G expressed predominantly cytosolic inactive BAX dimer, while HCT116 BAX KO reconstituted with BAX E75K expressed mostly cytosolic BAX monomer (Fig. 1f).

Moreover, western blot analysis confirmed that the generated cell lines HCT116 KO BAX WT, HCT116 KO BAX P168G, and HCT116 KO BAX E75K express similar levels of the major BCL-2 family proteins (Fig. 1g). Therefore, the generated HCT116 cell lines were selected to further investigate the role of the cytosolic inactive BAX dimer and monomer in apoptosis.

**Fig. 1** | **Diverse cancer cell lines express cytosolic inactive BAX monomers and BAX dimers. a** Schematic of the characterization of cytosolic BAX conformation in cancer cell lines. This figure was created with BioRender.com. **b** Size-exclusion chromatography (Superdex 200, HR 10/30 GL) of the cytosolic fraction of a diverse panel of cancer cell lines. Elution fractions were analyzed by western blot with a BAX antibody. The molecular weight as well as the BAX monomer and BAX dimer fractions are indicated. **c** Cellular localization of key BCL-2 family members in CALU-6 (left) and HCT116 (right). Cytosolic and mitochondrial fractions were separated, and samples analyzed by western blot using indicated antibodies. **d** Native blue gel of CALU-6 (left) and HCT116 (right) cytosolic fractions. The cytosolic fractions were treated with DMSO or BIM-SAHB ($50\,\mu M$) for 1 h on ice. The samples were analyzed by native blue gel and immunoblotted for BAX.

**e** Immunoprecipitation of active BAX in CALU-6 (top) and HCT116 (bottom). Cytosolic fractions failed immunoprecipitation by the 6A7 antibody. Treatment of cytosolic fractions with 1% triton exposed the 6A7 epitope on BAX and could be immunoprecipitated by the 6A7 antibody. **f** Size-exclusion chromatography (Superdex 200, HR 10/30 GL) of the cytosolic fraction of HCT116 BAX KO reconstituted with BAX WT (cytosolic dimer) or mutants, BAX P168G (cytosolic dimer) and BAX E75K (cytosolic monomer). Elution fractions were analyzed by western blot with a BAX antibody. **g** BCL-2 family proteins expression profile. Whole-cell lysates of HCT116 BAX KO reconstituted with BAX WT (cytosolic dimer) or mutants, BAX P168G (cytosolic dimer) and BAX E75K (cytosolic monomer) were analyzed by western blot with indicated antibodies. Blots are representative of at least n = 2 independent experiments. Source data are provided.

## The cytosolic inactive BAX dimer promotes resistance to apoptosis induction

We assessed the effect of ABT-263 (navitoclax), a clinical BH3 mimetic and inhibitor predominantly of BCL-XL and BCL-2, in cell viability of the HCT116 cell lines. HCT116 BAX E75K cells were the most sensitive with an $IC_{50}$ of 45 nM after 24 h. HCT116 BAX WT cells were four times less sensitive than the HCT116 BAX E75K monomer ($IC_{50}$ = 191 nM), and the HCT116 BAX P168G dimer cells were the most resistant to ABT-263 treatment. HCT116 BAX P168G cells did not show decreased viability even at the highest concentration of $2\,\mu M$ ABT-263 (Fig. 2a). We attributed this enhanced resistance to impaired mitochondrial translocation of the BAX P168G mutant[15,24]. Consistently, HCT116 BAX E75K cells showed more apoptosis compared to the HCT116 BAX WT cells as measured by annexin V staining. HCT116 BAX WT cells exhibited slightly more apoptosis than the HCT116 BAX P168G cells only at concentration of $1\,\mu M$ ABT-263 (Fig. 2b). Likewise, HCT116 BAX E75K demonstrated more potent caspase-3/7 activity than HCT116 BAX WT cells. Particularly, HCT116 BAX P168G cells did not show caspase-3/7 activity (Fig. 2c). Of note, we observed similar results in caspase-3/7 activation with a BCL-XL specific inhibitor A-1331852, whereas, a BCL-2 specific inhibitor ABT-199 failed to induce caspase-3/7 activity in all cell lines consistent with the cell lines expressing low levels of BCL-2 (Supplementary Fig. 3a, b, Fig. 1g). Importantly, doxorubicin, a chemotherapeutic drug that kills cells by mechanisms upstream of BCL-2 protein family, also induced more caspase-3/7 activation in HCT116 BAX E75K cells compared to HCT116 BAX WT cells (Fig. 2d). As with ABT-263, HCT116 BAX P168G cells were not responsive to doxorubicin at various concentrations (Fig. 2d).

From these results, we hypothesized that the expression of cytosolic inactive BAX dimer promotes resistance to apoptosis by impeding BAX activation by the various pro-apoptotic treatments. To test this hypothesis, we analyzed BAX activation using 6A7 immunoprecipitation. After a 2 h treatment with ABT-263, BAX E75K cells had more activated BAX than BAX WT cells, and we did not detect any active BAX in BAX P168G cells (Fig. 2e). Active BAX is established to translocate and form oligomers that create pores on the mitochondrial outer membrane through which cytochrome *c* exits. We used size exclusion chromatography to assess the translocation and oligomerization of BAX. At 6 h, most cytosolic BAX in the HCT116 BAX E75K cells had translocated to the mitochondria and formed oligomers (Fig. 2f). At the same time, HCT116 BAX WT cells showed a reduction of cytosolic BAX dimer into monomers, and some BAX translocation and oligomerization at the mitochondria which was considerably less compared to HCT116 BAX E75K cells (Fig. 2f). Like HCT116 BAX WT cells, cytosolic BAX dimers in HCT116 BAX P168G were disrupted into monomers, but no translocation was observed presumably due to impaired translocation of the BAX P168G mutant[15,24] (Fig. 2f). In agreement with the BAX translocation and oligomerization readouts, HCT116 BAX E75K cells showed substantially more cytochrome c release in the cytosol and mitochondrial depolarization compared to the HCT116 BAX WT cells (Fig. 2g, h). Consistent with the lack of BAX translocation, there were no detected changes in BAX P168G cells.

Altogether, our results suggest that the cytosolic inactive BAX dimer constitutes a roadblock to BAX activation and apoptosis induction by drugs that require BAX to induce apoptosis (Supplementary Fig. 3c).

## Discovery of a small molecule modulator of the cytosolic inactive BAX dimer

Considering the role of the cytosolic inactive BAX dimer in resistance to apoptosis in cancer cells, we sought to identify a small molecule that modulates cytosolic BAX dimer. First, we analyzed the interface between residues of the N-terminal trigger site interacting with residues of the C-terminal interface in the previously determined crystal structure of the autoinhibited BAX dimer (PDB 4SOO)[15] (Fig. 3a). We selected several residues to comprise a pharmacophore model hypothesis for in silico screening of small molecules. The pharmacophore model comprised several features: hydrophobic groups of Q28, L45, L47 and T172, aromatic group of F176, positively and negatively charged groups of R109 and D48 and hydrogen bond donor and acceptor groups of A46 and Y164 (Fig. 3b). An in silico library of ~14,000,000 diverse commercial compounds was screened to fit pharmacophore hypotheses using PHASE software (Supplementary Fig. 4a). The top 1000 ranked hits of the in silico screen were analyzed based on their molecular properties, fit to the pharmacophore model and predicted interactions with BAX. This led to the selection of 26 compounds, putative inactive BAX Dimer Modulators (BDM) with diverse structures.

BDM compounds were purchased and experimentally screened for disrupting the formation of cytosolic inactive BAX dimer of CALU-6 cells using the BN-PAGE assay (Fig. 3c, d, Supplementary Table 1). Furthermore, we used a previously established fluorescence polarization (FP) assay to identify small molecules that bind to BAX and compete binding of fluorescent-labelled BIM SAHB (FITC-BIM-SAHB) to the BAX trigger site[38]. BIM SAHB disrupted the cytosolic inactive BAX dimer of CALU-6 cells whereas BAX activator BTSA1 had no capacity to disrupt the cytosolic inactive BAX dimer (Fig. 3d, Supplementary Fig. 4b) despite its capacity to bind the BAX trigger site and compete FITC-BIM SAHB binding to BAX[39]. Interestingly, both screens identified three molecules BDM16, BDM17 and BDM19 as the most capable to disrupt the cytosolic inactive BAX dimer and also bind directly to BAX and compete FITC-BIM SAHB binding to BAX trigger site (Fig. 3d, e). Further analyses of the BDM16 and BDM17 chemical structures revealed that both molecules have a promiscuous binding core scaffold[40]. Therefore, we decided to further characterize BDM19 compound. We confirmed BDM19 identity by resynthesis and NMR analysis and measured an $IC_{50}$ value for competition with FITC−BIM SAHB in the FP assay (Fig. 3f). The molecular structure of BDM19 (molecular weight, 508 Da) is comprised of a quinazolinone core scaffold, substituted with a benzoic acid group, a methyl styryl group and an iodine (Fig. 3g).

## BDM19 binds at the N-terminal trigger site of BAX

To better understand the mechanism of interaction of BDM19 and its binding site with BAX, we attempted to determine its structure bound

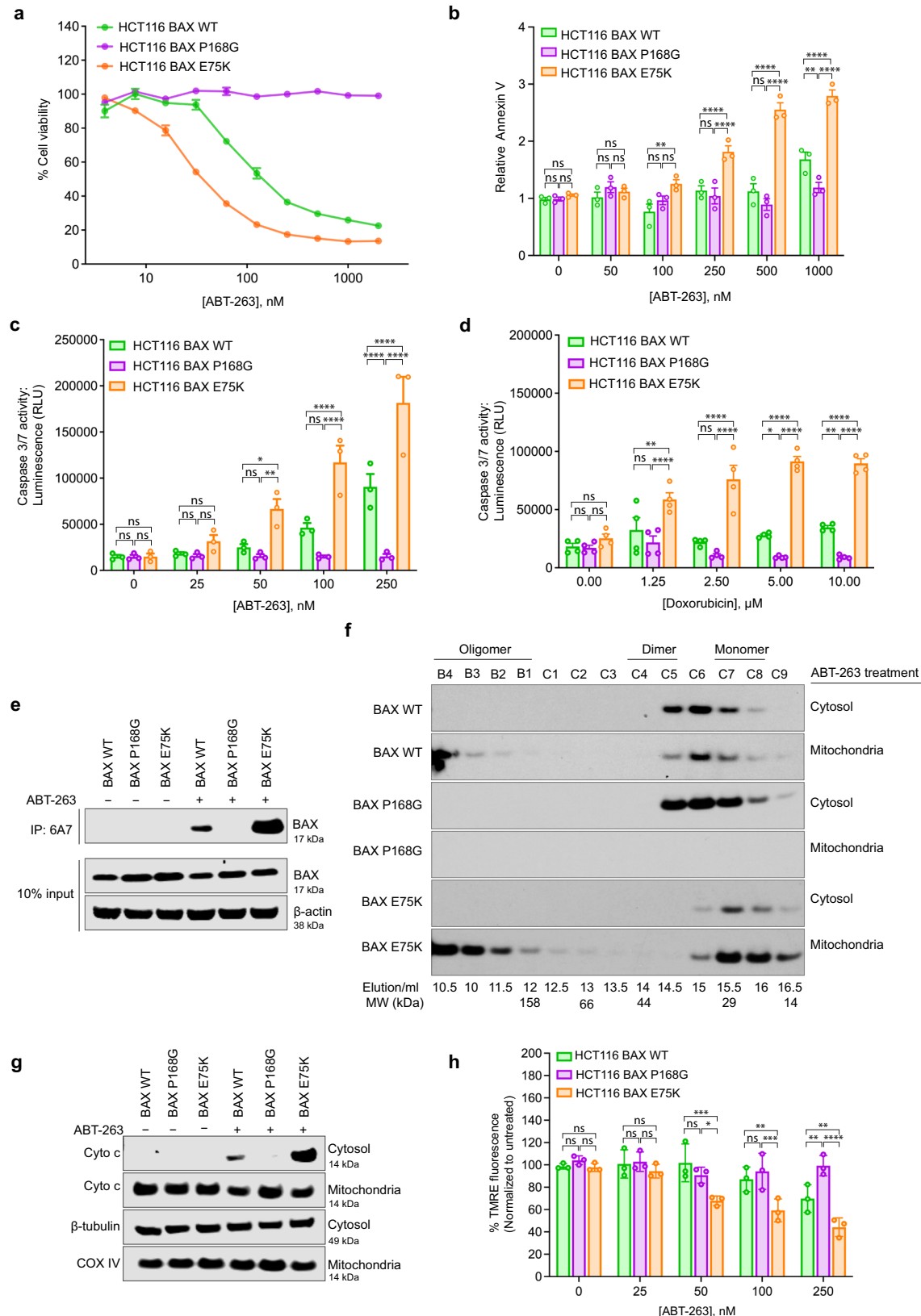

to BAX. We first performed crystallization trials to determine a co-crystal structure of BDM19 bound to BAX. However, determined BAX crystal structures lacked density of BDM19, perhaps due to increased conformational changes of BAX that are refractory to the crystal-lization process. Then, we performed 2D $^1$H–$^{15}$N heteronuclear single quantum coherence (HSQC) NMR analysis with $^{15}$N-labeled BAX as with

previous NMR studies of BAX[18,26,38,39,41]. Titration of BDM19 induced notable shifts of specific cross-peaks of BAX residues in the HSQC-NMR spectra. Several significant chemical shift perturbations (CSPs) were localized to the N-terminal BAX trigger site, including residues of α1 and α6 and α1-α2 loop (Fig. 4a, b). Additional CSPs corresponding to residues in adjacent helices to the trigger site, α2, α4 and α7 were also

**Fig. 2 | The cytosolic inactive BAX dimer promotes resistance to apoptosis induction. a** Cell viability (cell-titer glo assay) of HCT116 BAX KO reconstituted with BAX WT (cytosolic dimer) or mutants, BAX P168G (cytosolic dimer) and BAX E75K (cytosolic monomer) treated with a titration of ABT263 at 24 hrs. **b** Annexin V staining of HCT116 BAX KO reconstituted with BAX WT (cytosolic dimer) or mutants, BAX P168G (cytosolic dimer) and BAX E75K (cytosolic monomer) treated with a titration of ABT263 at 2 h**rs. c** Caspase-3/7 activity of HCT116 BAX KO reconstituted with BAX WT (cytosolic dimer) or mutants, BAX P168G (cytosolic dimer) and BAX E75K (cytosolic monomer) treated with a titration of ABT263 at 6 hrs. **d** Caspase-3/7 activity of HCT116 BAX KO reconstituted with BAX WT (cytosolic dimer) or mutants, BAX P168G (cytosolic dimer) and BAX E75K (cytosolic monomer) treated with a titration of doxorubicin for 24 hrs. **e** Active BAX immunoprecipitation with 6A7 antibody in HCT116 BAX KO reconstituted with BAX WT (cytosolic dimer) or mutants, BAX P168G (cytosolic dimer) and BAX E75K (cytosolic monomer) after 2 h treatment with vehicle (DMSO) or 250 nM of ABT263. **f** Translocation and oligomerization of BAX in HCT116 BAX KO reconstituted with BAX WT (cytosolic dimer) or mutants, BAX P168G (cytosolic dimer) and BAX E75K (cytosolic monomer) after 6 hrs treatment with 250 nM of ABT263. The cytosolic and mitochondrial fractions were separated and analyzed with size-exclusion chromatography (Superdex 200, HR 10/30 GL). **g** Cytochrome *c* release in HCT116 BAX KO reconstituted with BAX WT (cytosolic dimer) or mutants, BAX P168G (cytosolic dimer) and BAX E75K (cytosolic monomer) after 3 hrs treatment with vehicle (DMSO) or 250 nM of ABT263. **h** Mitochondria depolarization in HCT116 BAX KO reconstituted with BAX WT (cytosolic dimer) or mutants, BAX P168G (cytosolic dimer) and BAX E75K (cytosolic monomer). After a 6.5 h treatment with a titration of ABT263, mitochondria depolarization was assessed using tetramethylrhodamine (TMRE) staining. (**a**–**d**, **h**) are mean ± SEM for assays performed in triplicate and *n* = 3 independent experiments. (**e**–**g**) are representative of *n* = 3 independent experiments with similar results. Statistics were obtained using two-way ANOVA: ns, $p \geq 0.05$; *$p < 0.05$; **$p < 0.01$; ***$p < 0.001$; ****$p < 0.0001$. Source data are provided.

observed (Fig. 4a, b). Moreover, significant CSP was localized at the C-terminal surface including residues of helix α9 and α5 (Fig. 4a, c). Previous HSQC-NMR analyses of BAX activators such as BIM SAHB and BTSA1 have shown CSPs predominantly in trigger site residues of α1, α6 and α1-α2 loop but also highlighted allosteric CSP effects in α4, α7, and α9 that may be related to BAX activation process[19,38,39]. Crystal structures of BAX mutants e.g. W139A have also suggested that binding at the N-terminal trigger site may modulate local conformational changes at α4, α7, and α9[21]. BDM19 induced significantly more CSPs at residues of α9 compared to other molecules that have been studied by NMR and bind the BAX trigger site[18,26,38,39,41]. Because BDM19 was selected from a screen against a pharmacophore model that included residues of both the N-terminal surface and C-terminal surface of BAX, BDM19 may have the capacity to bind residues of the BAX dimer interface including those from the C-terminal interface[15].

Since both NMR and FP studies suggested binding of BDM19 to the BAX trigger site, we performed molecular docking of BDM19 to identify its predicted binding pose on the BAX trigger site using an induced-fit docking approach starting from the structure of the inactive BAX monomer[15,17]. As with Eltrombopag, a BAX trigger site inhibitor that possesses a carboxylic acid, BDM19 possess a carboxylic acid group as well[40]. We reasoned that the negatively charged carboxylate of BDM19 would form a favorable interaction with one of the three basic residues K21, R134, or R145 of the BAX trigger site and interestingly all these residues exhibited significant CSPs upon BDM19 binding (Fig. 4a). Therefore, docking was performed with a binding site to include these three residues but also with larger limits to account for potential ambiguity guided by the NMR data and consideration of alternative binding sites. We performed induced-fit docking (IFD) with the GLIDE software using a largely extended surface of BAX at the trigger site to exhaustively consider possible binding modes and the associated local conformational changes of α1, α6, and α1−α2 loop residues. The induced-fit docking calculations yielded several docking poses placing BDM19 at the juxtaposition of helices α1 and α6 surrounded by the sidechains of K21, R134 and R145 (Supplementary Fig. 5a). The majority and high-scoring poses of BDM19 show the carboxylate in interaction with the sidechain of K21 and there were lower scoring poses showing interaction with the sidechain of R134 and R145 (Fig. 4d, e, Supplementary Fig. 5b).

To further confirm the BDM19 pose in the BAX trigger site, we evaluated direct binding of BDM19 to BAX with microscale thermophoresis (MST), using a previously established BAX mutant (BAX 4C)[41,42] for MST studies, and additional mutations that reverse the charge of the basic trigger site residues: K21E, R134E, or R145E. These BAX mutants had similar purification and folding of monomeric BAX compared to WT by SEC as previously determined[41,42]. Direct binding of BDM19 to BAX was demonstrated with a calculated dissociation constant $K_D$ of 560 nM (Fig. 5a). Previously, we were only able to measure a stable binding isotherm by MST with two small molecule BAX inhibitors (BAI1 and Eltrombopag)[41,42]. In contrast, small-molecule BAX activators BTSA1 and BTSA1.2 do not generate a stable binding curve due to concurrently triggering BAX conformational changes that lead to BAX oligomerization[12,39]. BAX mutants exhibited a significant reduction in binding BDM19 with K21E ($K_D > 10000$ nM) having the biggest impact followed by R145E ($K_D = 9342$ nM) and R134E ($K_D = 4630$ nM), suggesting that K21 is most likely the residue forming a salt bridge with the carboxylate of BDM19 as suggested by docking studies (Fig. 5a). Interestingly, BAX K21E mutant reduced BAX activation induced by BAX activators BIM SAHB, BAM7 and BTSA1, however, it did not reduce inhibition in response to BAX inhibitor Eltrombopag[18,38,39,41]. Eltrombopag binding was impaired primarily with R145E mutant. Taken together, the data suggest that BDM19 has a specific and distinct interaction mode with the BAX trigger site among known small molecule trigger site binders.

Next, we analyzed the most energetically favorable BDM19 pose from docking, which had the negatively charged carboxylate on the benzoic acid making a salt bridge with K21 (Figs. 4d and 5b, c). In this pose, the benzene ring makes hydrophobic interactions with adjacent residues A24 and L25. The carbonyl group of the quinazolinone forms electrostatic interactions with a side chain of Q28. R134 and R145 form cation π-stacking interactions with the benzene rings on the quinazolinone and the toluene groups, respectively (Fig. 5b,c). Moreover, quinazolinone and the toluene groups have a number of van der Waals contacts with hydrophobic residues M137, L141 and M20 (Fig. 5b, c). This particular pose is consistent with impaired binding from the BAX mutants K21E, R134E, R145E (Fig. 5a). To further probe this BDM19 pose, we tested binding with the Q28A mutant that should reduce the interaction with the carbonyl group of quinazolinone. Indeed, this mutant abolished the interaction with BDM19 and BAX ($K_D > 10000$ nM) (Fig. 5a, b). Furthermore, to probe the specificity of BDM19 and the impact of its predicted charged interaction with K21, we utilized a BDM19 analog featuring an ethyl ester (BDM19.1) in place of the carboxylic acid (Supplementary Fig. 5c). The addition of an ethyl group eliminates the anionic carboxylate and BDM19.1 exhibited a dramatically diminished binding affinity (Fig. 5a), highlighting the necessity of the carboxyl group. We also generated additional analogs to replace the phenyl of the styryl group with non-aromatic and less bulky cyclopentyl and cyclopropyl groups of BDM19.2 and BDM19.3, respectively (Supplementary Fig. 5c). Moreover, BDM19.4 reduced an aromatic phenyl ring from the quinazolinone ring of BDM19. These three analogs had also drastically decreased binding affinity to BAX compared to BDM19 (Fig. 5d). In addition, when all analogs tested for BAX activation in CALU-6 cells expressing cytosolic BAX dimers, only BDM19 induced BAX activation as probed with 6A7 antibody co-immunoprecipitation (Fig. 5e). These data suggest that interactions predicted for BDM19 with R145 and R134 residues, in addition to the

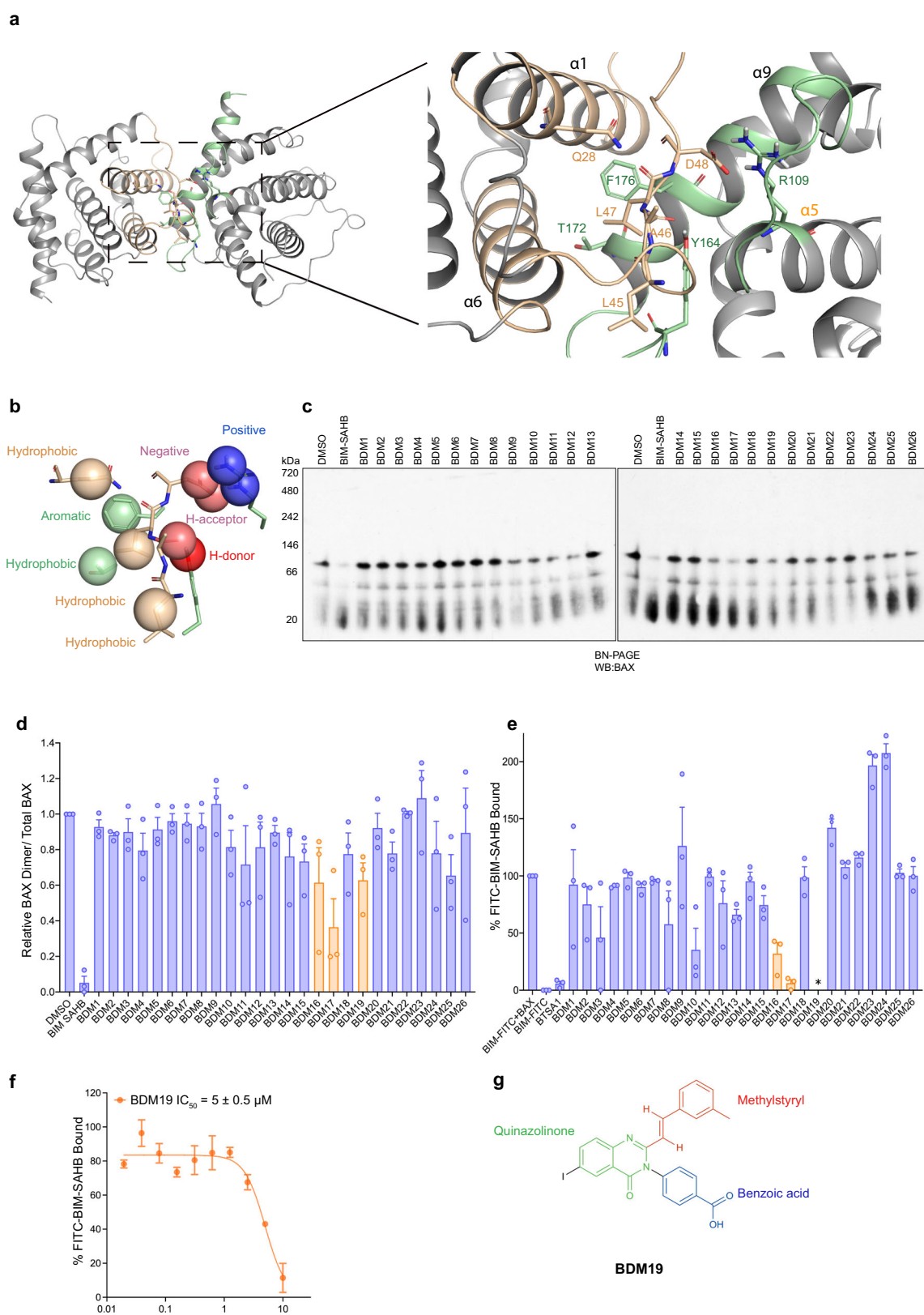

interaction with K21, contribute to binding and activating BAX. Consistently, BDM19 binds only to inactive BAX and not to detergent-activated BAX (Supplementary Fig. 5d).

Finally, we evaluated the activity of BDM19 with cytosolic BAX using *Bax* and *Bak* double knockout (DKO) MEFs reconstituted with BAX WT and mutants BAX K21E, R134E and R145E. Interestingly, using

the BN-PAGE assay, cytosolic mutant BAX R134E and BAX R145E proteins express as BAX monomers (Supplementary Fig. 6a). Not unexpectedly, both R134 and R145 form interactions at the inactive BAX dimer interface[15], suggesting that the reverse charge mutations disrupt the cytosolic BAX dimer to monomer. Despite that BDM19 and BIM SAHB were effective to reduce cytosolic BAX WT dimer to active

**Fig. 3 | Discovery of a small molecule modulator of the cytosolic inactive BAX dimer. a** Ribbon representation of the dimer interface of the autoinhibited BAX dimer structure (PDB: 4S0O) **b** Pharmacophore model based on residue sidechains comprising hydrophobic, aromatic, hydrogen bond donor or acceptor, positive or negative charge groups for the in silico screen. **c** Functional screen using BN-PAGE. The cytosolic fraction of CALU-6 was incubated with DMSO, 50 μM BIM SAHB or 50 μM of compounds for 1 h at 30 °C and on ice for BIM SAHB. **d** Ratio of cytosolic BAX dimer over total cytosolic BAX in (**c**). **e** Competitive fluorescence polarization binding assay of compounds at 10 μM using fluorescent-labeled BIM SAHB (FITC-BIM SAHB) bound to BAX. **f** Competitive fluorescence polarization binding assay of BDM19 using FITC-BIM SAHB bound to BAX. **g** Chemical structure of BDM19. (**c**) is representative of three independent experiments with similar results. **d**–**f** Data are mean ± SEM from *n* = 3 independent experiments. * denotes values <0. Source data are provided.

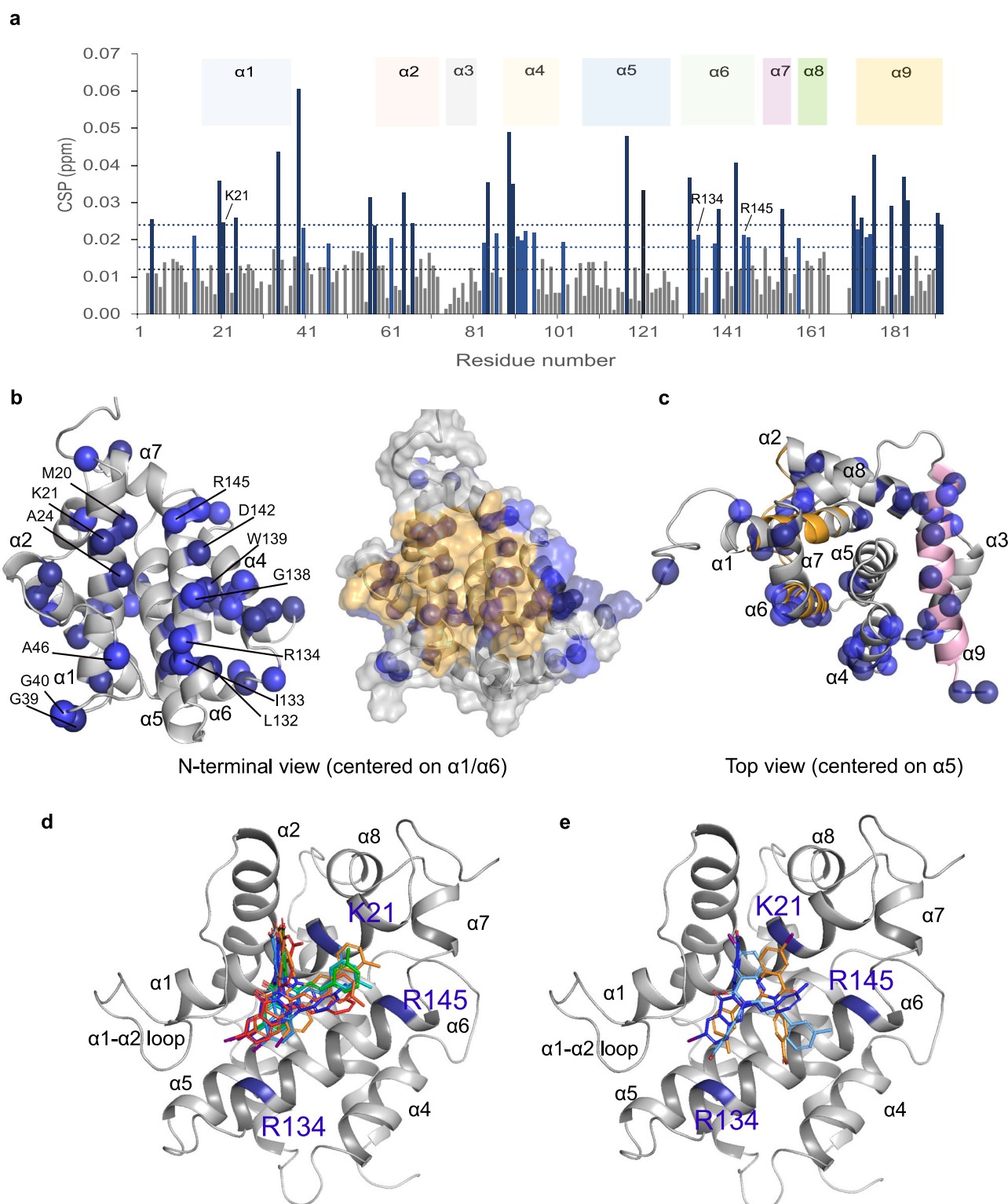

**Fig. 4 | BDM19 binds the BAX trigger site. a** Chemical shift perturbations (CSPs) of $^{15}$N-labeled BAX measured in the presence of 1:2 BAX:BDM19 compared to unbound BAX are plotted as a function of BAX residue number. Residues with chemical shift perturbations over the significance threshold or two times the significance threshold are labeled light blue or dark blue, respectively. The black dotted line represents the average CSP. Residues that are part of the nine helices of the BAX structure are designated with a colored shading. Data are representative of $n = 3$ independent experiments. **b** Mapping of residues undergoing significant CSPs to the ribbon (left) and the transparent surface (right) structure of BAX (PDB: 1F16). Residues with significant CSPs on the trigger site are labeled. View of the N-terminal trigger site (orange surface) of BAX showing residues with significant CSPs clustered on the trigger site surrounding a hydrophobic pocket formed by α1 and α6. **c** Mapping of residues undergoing significant CSPs to the ribbon structure of BAX (PDB: 1F16) as in (**b**) with a top view of the BAX structure showing significant CSPs in α4, α5, α7 and α9 (pink). α1 and α6 are shown in orange. **d** Induced-fit docking of BDM19 into the BAX structure (PDB: 1F16) guided by the NMR mapping data. Ribbon structure of BAX with 7 BDM19 poses (sticks in different colors) clustering into a similar site with different orientations based on the interaction of BDM19 with the basic residues K21, R134, R145 of the N-terminal trigger site. **e** Ribbon structure of BAX with 3 representative BDM19 poses (sticks in different colors) with different orientations based on the interaction of BDM19 with the basic residues K21, R134, R145 of the N-terminal trigger site. Source data are provided.

monomer both compounds had diminished activity on the BAX R134E and BAX R145E cytosolic monomers (Supplementary Fig. 6a). In line with these results, BDM19 significantly decreased the melting point of BAX WT by ~4.5 °C compared to the DMSO control, consistent with direct engagement of cellular BAX (Supplementary Fig. 6b). However, BDM19 had significantly diminished capacity to target any of the BAX K21E, R134E and R145E mutants (Supplementary Fig. 6c-e). Consistent with these results, BDM19 in combination with a sublethal dose of ABT-263 induced apoptosis in DKO MEFs expressing BAX WT and BAX activation as probed with 6A7 antibody co-immunoprecipitation (Supplementary Fig. 7a, b). However, BDM19 was not as effective to induce apoptosis or activate BAX in MEFs expressing BAX K21E, R134E and R145E mutants. Taken together, the data demonstrate that BDM19 binds directly to BAX with a distinct binding interaction to the BAX trigger site that induces cytosolic BAX dimer activation.

## BAX conformational modulation by BDM19

Since the N-terminal BAX trigger site has been established as the binding site for BAX activators and the BAX inhibitor Eltrombopag, we set out to understand the effect of BDM19 on the conformation of BAX and how it can modulate the cytosolic BAX dimer. We performed three independent molecular dynamics (MD) simulations of 1μs for the unbound inactive BAX and the BAX-BDM19 complex structures that showed reasonable equilibration (Supplementary Fig. 8a-d). The overall structure of BAX was maintained but undergone larger fluctuations in BAX-BDM19 complex structures compared to unbound BAX, as determined by the root mean square deviation (RMSD) (Supplementary Fig. 8a). The distances between the carboxylate and carbonyl of BDM19 with residues K21 and Q28, respectively are more stable throughout the simulations (Fig. 6a, b, d). The distances between BDM19 toluene and iodobenzene with residues R145 and R134 respectively were more dynamic but remained in close proximity (Fig. 6a, c, e). These BAX-BDM19 distances strongly correlate with the mutagenesis and binding data supporting that BDM19 interacts with K21 and Q28 and to a lesser strength with R134 and R145 (Figs. 5a, 6a).

Mobilization of the α1–α2 loop from a closed conformation to a more open conformation is one of the conformational changes in the activation process of BAX[14,15,19]. HSQC CSP suggested BDM19 binding directly affected residues of the α1–α2 loop. In the unbound inactive BAX structure, residues P49 and D48 on the α1–α2 loop are in the vicinity of L141 and D142 on α6, respectively. In the BAX-D19 complex simulations, the distance between P49 and L141 as well as the distance between D48 and D142 displayed a slight increase (Supplementary Fig. 8e-h). Moreover, the root mean square fluctuation (RMSF), which is a measure of the dynamics of each residue in the BAX structure, showed an increase in dynamics of α1–α2 loop residues adjacent to the BH3 domain (Supplementary Fig. 8i). Furthermore, RMSF analysis showed significant changes in two other regions. The α4–α5 loop and the N-terminal of α7 displayed a decrease in RMSF while α3–α4 loop and the C-terminal helix α9 displayed an increase in RMSF (Supplementary Fig. 8i). Allosteric communication between the N-terminal trigger site and the C-terminal canonical site has been previously described to be mediated by interactions of the α4–α5 loop and α7[19,21,41]. Thus, we measured the distance between residues N104 and F105 on the α4–α5 loop as well as W151 and Q155 on α7 and observed an increase in distances for all the residues in the BDM19-BAX structures compared to unbound BAX structures (Fig. 6f, Supplementary Fig. 9c-j). These data agree with the proposed mechanism that binding to the N-terminal trigger site allosterically transmits conformational changes throughout BAX[19,21]. The dissociation of the α9 helix from the canonical site is required for BAX translocation to the mitochondria, and the α3–α4 loop and α9 interface comprise the opening of the canonical site[19,21,41]. To evaluate the effect of BDM19 to this key hotspot of the canonical site, we measured the distances between four residues M79, T85, V91 and K189 (Fig. 6g, Supplementary Fig. 10c-i). Most distances between the indicated residues were increased. To quantify the opening of the canonical site, we calculated the area of the site considering the canonical site as two triangles. The canonical site area of unbound BAX was 96 Å$^2$ and of the BAX-BDM19 complex was 104.5 Å$^2$, an increase of ~9% (Fig. 6g). Taken together, the MD data suggests that binding of BDM19 at the N-terminal trigger site induces direct changes at the trigger site (α1–α6 interface) and distal conformational changes primarily at the α4–α5 loop–α7 interface and α3–α4 loop–α9 interface. These changes align with the BAX activation mechanism of the inactive BAX structure, supporting an allosteric activity of BDM19 that couples the trigger site and α9 at the canonical site.

To compare BDM19 with another BAX activator, we performed three independent molecular dynamics (MD) simulations of 1μs for the BAX-BTSA1 complex under the same conditions (Supplementary Fig. 8a-d). BTSA1 also induced changes to the trigger site and further increased the distances with the α1–α2 loop residues and α6 residues (P49-L141, D48-D142) (Supplementary Fig. 8e-h). BTSA1 also induced distal conformational changes at the α4–α5 loop–α7 interface (Supplementary Fig. 9a-j) and α3–α4 loop - α9 interface (Supplementary Fig. 10a-l). Overall, BTSA1 in comparison to BDM19 affected the same residues and interfaces but in several measured distances BTSA1 induced larger changes, which may suggest a better potency for BTSA1 to activate the BAX monomer than BDM19. Interestingly, Eltrombopag, a small molecule BAX inhibitor that binds at the N-terminal trigger site and induced stabilization of the inactive BAX structure and inhibition of BAX activation[41], induced the opposite effects of BDM19-BAX complex. Specifically, binding of Eltrombopag at the trigger site induced a decrease in distances among residues of the α4–α5 loop and α7 and a decrease in dynamics and distances among residues at the α3–α4 loop and the C-terminal α9 helix[41].

## BDM19 potentiates BAX-mediated apoptosis

To evaluate the cellular activity of BDM19, we tested its ability to promote apoptotic cell death in various lymphoma and leukemia cell lines expressing either cytosolic BAX monomer (e.g. HPB-ALL) or dimer (e.g. SUDHL-5) (Fig. 1b). BDM19 decreased cell viability in all cell lines and promptly induced caspase-3/7 activity at 6 hrs (Fig. 7a, Supplementary Fig. 11a). Using SUDHL-5, the most sensitive cell line to BDM19 in cell viability (IC$_{50}$ = 1.36 μM), we observed increase in caspase-3 and PARP cleavage, characteristic of apoptosis induction at

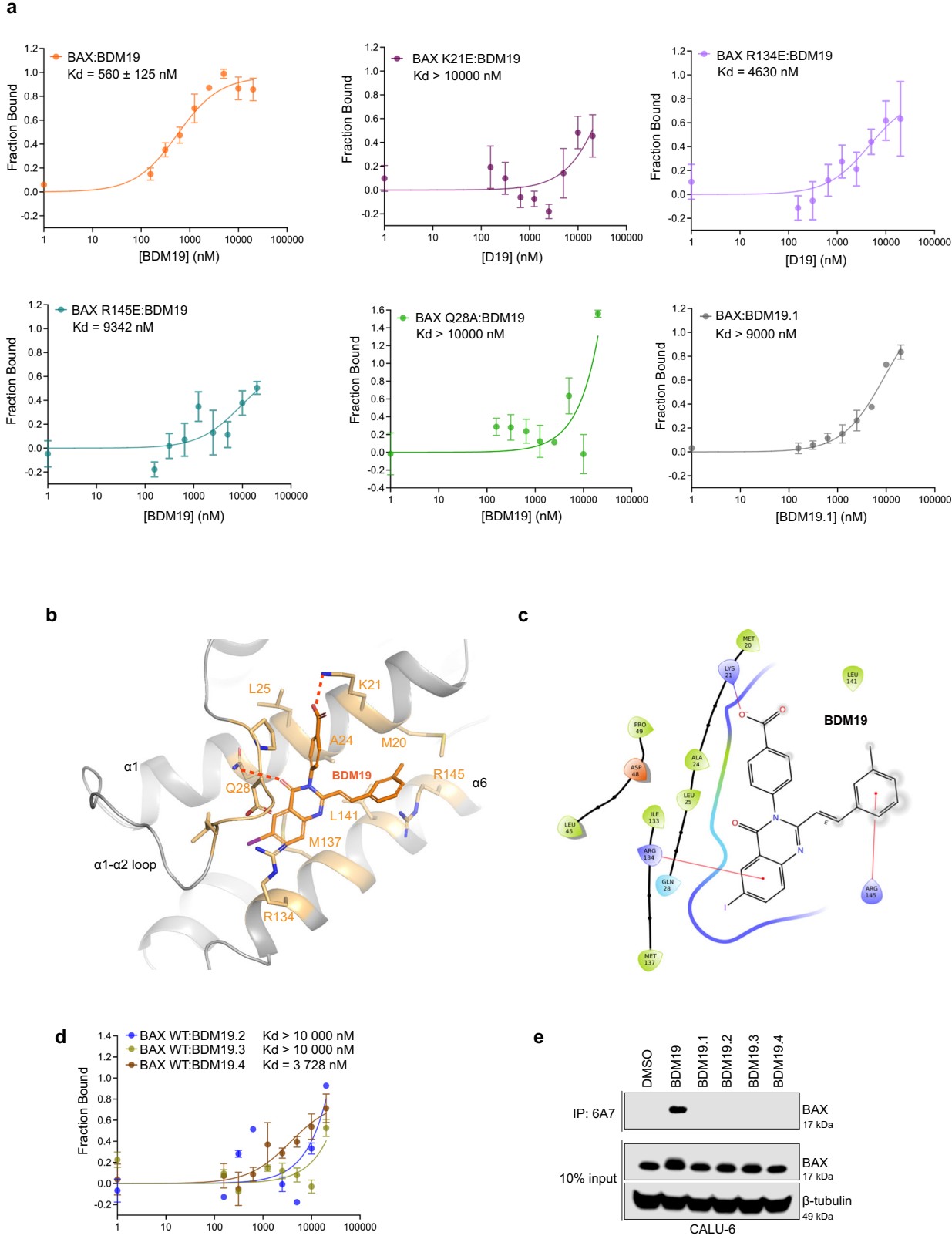

6 hrs (Fig. 7b, c). Moreover, in SUDHL-5 cells, we evaluated BAX mitochondrial translocation and cytochrome c release from mitochondria into the cytosol by separating cytosolic and mitochondrial fractions. Indeed, BDM19 induced dose-responsively BAX mitochondrial translocation and cytochrome c release at 6 hrs (Fig. 7d). Overall, these results suggest that BDM19 has capacity to induce BAX activation

and apoptosis as a single-agent treatment in cell lines expressing both cytosolic BAX monomer and dimer.

Since previously established BAX activator BTSA1 showed no capacity to activate the cytosolic inactive BAX dimer (Supplementary Fig. 4b), we compared its activity with BDM19 in cell lines expressing cytosolic BAX dimer (CALU-6) and cytosolic BAX monomer

**Fig. 5 | BDM19 binds the trigger site through interactions with K21, Q28, R134 and R145. a** Microscale thermophoresis direct binding of BDM19 to BAX-4C (orange), and BAX-4C single mutants K21E (jam), R134E (lavender), R145E (teal) and Q28A (green) and of an ethyl-ester analog BDM19.1 to BAX-4C (grey). Data are mean ± SEM from n = 3 independent experiments. **b** Close up view of the lowest energy docked pose of BDM19 consistent with NMR and MST studies. **c** BAX-BDM19 interaction cartoon from (**b**) showing key hydrophobic contacts, hydrogen bonds and cation-π interactions of BMD19. **d** Microscale thermophoresis direct binding of BDM19 analogs BDM19.2 (blue), BDM19.3 (mustard) and BDM19.4 (brown) to BAX-4C. **e** Immunoprecipitation of active BAX in CALU-6. CALU-6 cells were treated with DMSO or 20 μM BDM19 or its analogs BDM19.1, BDM19.2, BDM19.3 and BDM19.4 for 2 hrs. Only BDM19 induced the exposure of the 6A7 epitope on BAX and BAX could be immunoprecipitated by the 6A7 antibody. **a**, **d** Data are mean ± SEM from n = 3 independent experiments. (**e**) are representative of at least n = 2 experiments. Source data are provided.

(OCI-AML3). We also used these molecules in combination treatment with ABT-263 since we previously found that BAX activation has synergistic activity with inhibition of BCL-XL or BCL-2 by ABT-263[12]. BDM19 and BTSA1 had similar weak activity in cell viability in both cell lines as single-agent treatments (Fig. 7e). However, BDM19 strongly synergized with ABT-263 compared to BTSA1 in CALU-6 cells (Fig. 7e, f). In contrast, in OCI-AML3 cells, we did not observe synergy between BDM19 with ABT-263, except at high concentrations (Fig. 7e, f). However, we observed a strong synergy between BTSA1 and ABT-263 in OCI-AML3 cells (Fig. 7e, f). To decipher the underlying mechanism of these results, we evaluated BAX activation upon the combination of BDM19 or BTSA1 with ABT-263 in CALU-6 and OCI-AML3 cells, using the 6A7 antibody, which recognizes only activated BAX. In CALU-6 cells, the BDM19 and ABT-263 combination induced more potently BAX activation than the BTSA1 and ABT-263 combination (Fig. 7g). In contrast, in OCI-AML3 cells, we observed significant BAX activation by the BTSA1 and ABT-263 combination, while there was minimal BAX activation by the BDM19 and ABT-263 combination (Fig. 7g).

Next, we confirmed the BAX-dependent specificity of BDM19's pro-apoptotic activity and synergistic activity with ABT-263. BDM19 potentiated apoptosis with ABT-263 as increased caspase-3/7 activation was observed with the combination compared to ABT-263 alone in CALU-6 (Fig. 7i). However, in CALU-6 cells with BAX knockout (CALU-6 BAX KO) no caspase-3/7 activation was observed by either BDM19 or ABT-263 or their combination (Fig. 7i). In CALU-6 cells with BAK knockout (CALU-6 BAK KO), BDM19 potentiated apoptosis with ABT-263 similarly to wild type CALU-6 cells but to a lesser degree, suggesting a contribution of BAK in apoptosis of CALU-6 cells by BDM19 and ABT-263 supposedly through interactions with activated BAX and formation of BAX-BAK oligomers (Fig. 7i). Taken together, these results are consistent with a specific activity of BDM19 on cytosolic inactive BAX dimer that leads to BAX activation and BAX-mediated apoptosis (Fig. 8a). Lastly, we found that BDM19 also potentiates apoptosis with doxorubicin in HCT116 BAX KO cells reconstituted with BAX WT (cytosolic dimer) and with mutant BAXP168G (cytosolic dimer) (Supplementary Fig. 11b, Fig. 1f), suggesting a utility of BDM19 with BAX mutants resistant to BAX activation and apoptosis.

## Discussion

Understanding BAX and BAK activation mechanisms and their regulation has been portrayed as the "holy grail" of apoptosis research because of their fundamental importance to initiate and commit the cell to mitochondrial dysfunction and apoptosis, but also because of the complexity in elucidating these mechanisms. Here, our systematic analysis of cytosolic fractions of various hematological and solid tumor cell lines supports the existence of cytosolic inactive BAX dimers in several cancer cells. Comparatively from our collection of cell lines, leukemia and lymphoma cell lines have more cytosolic monomers than solid tumor cell lines and non-cancerous cells. Interestingly, our data link BAX dimers with higher expression levels of BAX and less priming of cells to mitochondrial apoptosis. This suggests that BAX dimers may be favored because of low levels of activated BH3-only proteins in low-primed cells, whereas in higher primed cells, higher levels of activated BH3-only proteins may promote the transition of BAX dimers to monomers. Because of the different mutational landscape and expression levels of proteins that regulate apoptosis

e.g. BCL-2 proteins, we evaluated the impact of cytosolic inactive BAX dimer and monomer in sensitivity or resistance to apoptosis using HCT116 isogenic cell lines expressing either BAX dimer or monomer in the same cellular background, using previously characterized mutations regulating the BAX dimer formation[15]. Our findings show that cells with cytosolic BAX dimer were more resistant to apoptosis and showed less BAX activation in response to BH3 mimetics and doxorubicin, whereas cells with cytosolic BAX monomer were more sensitive to apoptosis and showed more BAX activation. Therefore, these data suggest that the formation of inactive BAX dimers is an additional regulatory mechanism adopted by cancer cells to control apoptotic priming, besides the overexpression of anti-apoptotic BCL-2 proteins and suppression mechanisms of BH3-only proteins[9,11,12].

Although mutations of BAX that can promote its inactivation are not found frequently in cancer cells, more than 40 single residue mutations have been identified in cell lines and patient samples from available sequencing datasets (cBioPortal.com)[34,43–45]. While we have shown that two of these mutations G67R and P168G in BAX can form the inactive BAX dimer structure and are associated with resistance to BAX activation and apoptosis, our panel of cell lines do not possess mutations in BAX[15,34,45]. It is interesting though that cancer cells have mostly mutations in BAX on residues of the trigger site (e.g. A24V, A42S, T135P) and the C-terminal α9 (e.g. A178V, S194I)[44]. Although such mutations could modulate the process of BAX activation, it is reasonable to suggest that they may change the cytosolic BAX conformation by promoting or preventing stabilization of the inactive BAX dimer structure[15,18,21]. Besides BAX mutations, phosphorylation has been described in specific BAX residues such as S184 through AKT kinase or T167 through JNK and p38 kinases in cancer cells[28,46,47]. These modifications can promote or inhibit translocation of BAX from cytosol to the mitochondria and BAX activation. Since these residues are part of the inactive BAX dimer interface[15], they may also promote the formation of the inactive BAX dimer. Despite this possibility, our proteomic analysis in cell lines expressing cytosolic inactive BAX dimer or monomer failed to identify post-translational modifications on BAX. Moreover, our data here excluded the interaction of the cytosolic BAX dimer with anti-apoptotic BCL-2 proteins, consistent with previous studies[15,16]. In addition, previous proteomic studies have not identified proteins bound to the inactive cytosolic BAX[16]. Nevertheless, additional mechanisms related to the role of lipids or proline cis-trans isomerization have been suggested to modulate BAX and may also modulate the cytosolic BAX dimer/monomer ratio, however, these mechanisms are challenging to detect in vivo[48,49].

To further study the functional role of the inactive BAX dimer, we considered to identify a chemical probe that would induce activation of cytosolic inactive BAX dimers. Our pharmacophore-based in silico screening of ~14 million compounds using the previously described crystal structure of inactive BAX dimer coupled with BN-PAGE and FPA screening assays, yielded BDM19, which binds BAX and activates the cytosolic inactive BAX dimer. Our NMR, molecular modeling, mutagenesis and binding studies showed that BDM19 can engage the surface of the N-terminal trigger site of BAX using hydrophobic contacts with hydrophobic residues in the pocket of the α1, α6 and α1-2 loop, a salt-bridge with K21, a hydrogen bond interaction with Q28 and cation-π stacking interactions with R134 and R145. This BDM19 binding mode is distinct among other trigger site binders, the BAX activator BTSA1

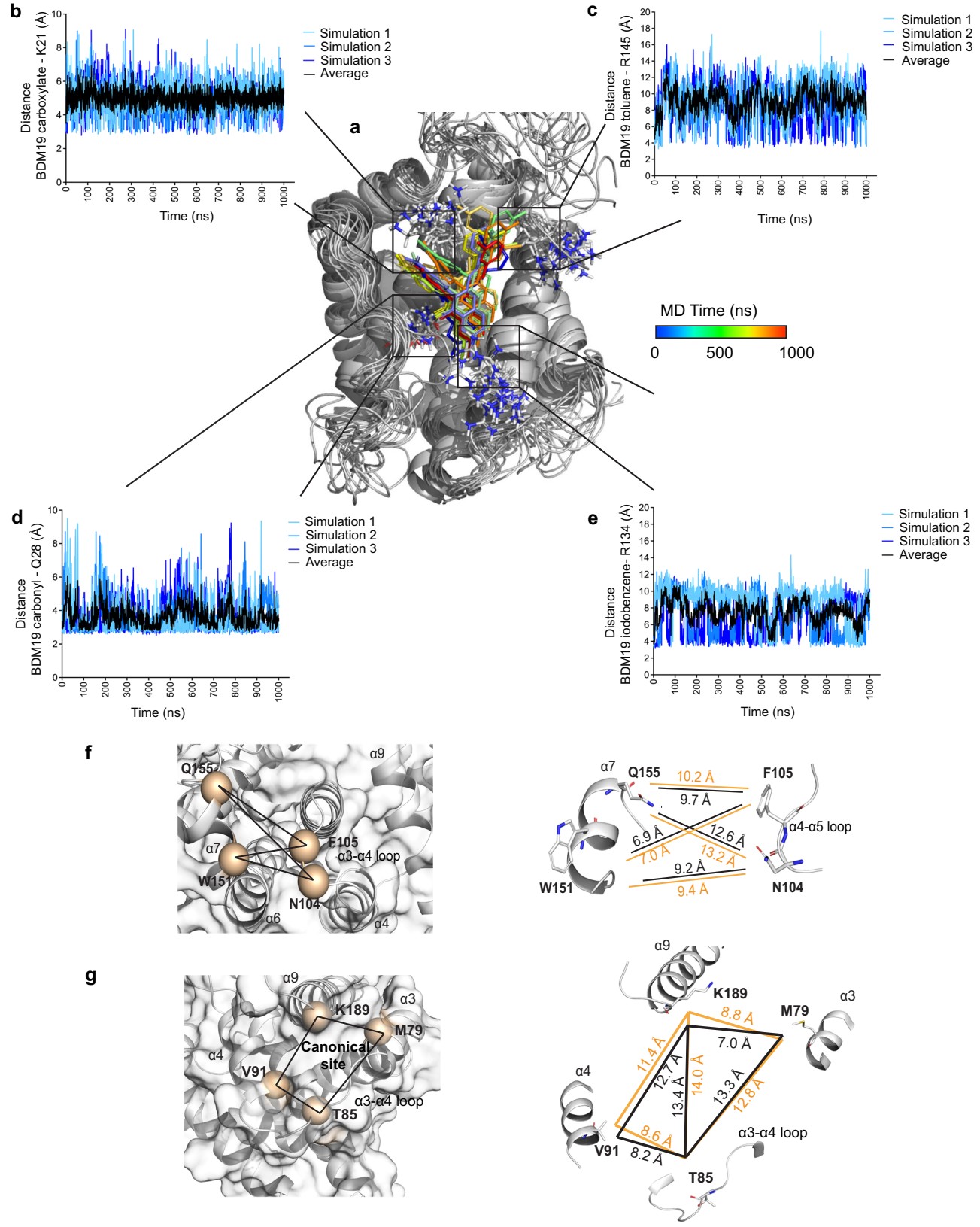

and the BAX inhibitor Eltrombopag (Fig. 8b)[39,41]. Interestingly, BDM19 is localized centrally to the juxtaposition of helices α1 and α6 having common interactions with BTSA1 and BAX residues e.g. K21, A24, Q28 in α1 and common interactions with Eltrombopag and BAX residues e.g. R134, M137, R145) in α6. Furthermore, NMR and molecular dynamics data suggested that BDM19 has allosteric conformational effects primarily at the α4–α5 loop - α7 interface and α3–α4 loop - α9 interface, consistent with the allosteric coupling of the trigger site and α9 at the canonical site, and supporting the BAX conformational activation process[19,21,39,41]. This distinct binding mode of BDM19, between BTSA1[39] and Eltrombopag[41], could explain the capacity of BMD19 to activate the inactive BAX dimer but having reduced potency

**Fig. 6 | Trigger site binding of BDM19 allosterically changes BAX conformation.**
**a** Overlay of structures of BAX-BDM19 complex from every 100 ns intervals from 0 to 1000 ns of molecular dynamics (MD) simulations. BDM19 color spectrum corresponds to time as described, BAX ribbon structure is colored gray with residues of interest represented as sticks for clarity. **b** Distance relative to time of BDM19 carboxylate-K21(ζ nitrogen). **c** Distance relative to time of BDM19 toluene (carbon-1)-R145 (ε nitrogen). **d** Distance relative to time of BDM19 carbonyl-Q28 (ε nitrogen). **e** Distance relative to time of BDM19 iodobenzene (carbon-6)-R134 (guanidine nitrogen NH₂). **b**–**e** Different shades of blue represent individual MD simulation distances, and black represents mean of $n = 3$ simulations. **f** Changes in structure and dynamics of the α7/α4-α5 loop interface. Transparent surface with ribbon representation of α7/α4-α5 loop interface with main residues highlighted in orange (left), and graphical representation of distances between residues at the α7/α4-α5 loop interface (right). Distances for BAX and BAX-BDM19 complex MD simulations are represented in black and orange respectively. **g** Changes in structure and dynamics of the canonical site opening formed by α3, α3-α4 loop, α4, and α9. Transparent surface with ribbon representation of the canonical site opening with main residues highlighted in orange (left), and graphical representation of distances between residues at the canonical site opening (right). Distances for BAX and BAX-BDM19 complex MD simulations are represented in black and orange respectively. **f**, **g** Distances represent the mean difference of $n = 3$ BAX and BAX-BDM19 MD simulations. Source data are provided.

to activate the BAX monomer. In contrast, BTSA1 binding characterized primarily by the interaction with K21 and not by the other two charged residues R134 and R145, has higher potency to activate the BAX monomer, but not the inactive BAX dimer (Fig. 8b)[39].

In summary, our findings here provide insights into the regulation of BAX through its cytosolic conformation and advance our understanding of mechanisms of resistance to BAX-mediated apoptosis in cancer cells. Furthermore, the discovery of BDM19 offers a chemical probe for investigating the role of cytosolic inactive BAX dimers (wild type or mutant) in cancer and potentially other homeostatic and pathological cellular contexts. BDM19 has distinct properties and a unique scaffold among the previously described BAX binders of the trigger site[14], offering an opportunity for the development of a new class of drugs to induce BAX activation and BAX-mediated apoptosis in cancer and other diseases.

## Methods

### Compounds
Hydrocarbon-stapled peptide corresponding to the BH3 domain of BIM, FITC-BIM SAHB: (FITC-βAla-EIWIAQELRS5IGDS5FNAYYA-CONH2) where S5 represents the non-natural amino acid inserted for olefin metathesis, was synthesized, purified at >95% purity by CPC Scientific Inc[38]. Compound BDM19 (Cat. # STK995952) and BDM19.1 (Cat. #STK549183) were provided by Vitas-M Laboratory. Navitoclax/ABT-263 (Cat. #S1001), Venetoclax/ABT-199 (Cat. #S8048) and A-1331852 (Cat. # S7801) were provided by SelleckChem. Doxorubicin hydrochloride (Cat. #D1515-10MG) was provided by Sigma. BTSA1 was synthesized and characterized (>98% purity) as previously described[39]. Synthesis of BDM19, BDM19.2, BDM19.3 and BDM19.4 and intermediates are described in Supplementary Methods. All NMR spectra are shown in Supplementary Figs. 12–16. All compounds were >95–98% pure and were reconstituted in 100% DMSO to prepare a 10 mM stock solution except for BDM19.1 (2.5 mM) and BDM19.4 (5 mM) and diluted in aqueous buffers or cell culture medium for assays.

### Cell Lines
Cell lines were purchased from ATCC and DSMZ. Leukemia cells: OCI-AML3 (DSMZ Cat. # ACC-582), U937 (ATCC Cat. # CRL-1593.2), NB4 (DSMZ Cat. # ACC 207), MOLM13 (DSMZ Cat. # ACC 554), HPB-ALL (DSMZ Cat. # ACC 483). Lymphoma cells: SUDHL-5 (ATCC Cat. # CRL-2958), Namalwa (ATCC Cat. # CRL-1432), SUDHL-16 (ATCC Cat. # CRL-2964). Non-small lung cancer cells: CALU-6 (ATCC Cat. # HTB-56). Colorectal cancer cells: COLO-320 (DSMZ Cat. # ACC 144), DLD1 (ATCC Cat. # CCL-221), RKO (ATCC Cat. # CRL-2577), HT29 (ATCC Cat. # HTB-38). HCT116 and HCT116 BAX KO were provided by the Bert Vogelstein laboratory. HCT116 BAX WT, HCT116 BAX P168G, HCT116 BAX E75K, MEF BAX WT, MEF BAX K21E, MEF BAX R134E and MEF BAX R145E stable cell lines were generated using retroviral transduction of HCT116 BAX KO and MEF BAX BAK DKO respectively with BAX-IRES-GFP. Non-cancerous cells: BEAS-2B (ATCC Cat. # CRL-3588) and IMR90 (ATCC Cat. # CCL-186). Cell lines were authenticated by their vendor and the Albert Einstein College of Medicine Genomics Core Facility. Morphology, karyotyping, and STR profiling were performed to confirm the identity of human cell lines and to rule out both intra- and inter-species contamination.

### Subcellular fractionation
To isolate the cytosolic and mitochondrial cellular fractions, cells were lysed by a Dounce homogenizer in a mitochondria isolation buffer (IBc) (10 mM Tris HCl, 1 mM EGTA, 200 mM sucrose, pH 7.5) supplemented with Halt protease inhibitor cocktail (Thermofisher Cat. 1861279). Cell lysates were centrifuged at 700 $g$ for 10 min at 4 °C to remove non-lysed cells and nuclei. The supernatant was centrifuged at 12,000 $g$ for 10 min at 4 °C, the supernatant was collected as the cytosolic fraction and the resulting pellet as the mitochondrial fraction. The membrane pellet was resuspended in IBc buffer containing 0.5% CHAPS, incubated for 60 min on ice and centrifuged at maximum speed for 10 min to collect solubilized membranes.

### Size-exclusion chromatography analysis
Superdex 75 10/300 GL and 200 10/300 GL columns (Cytiva) were used for size exclusion chromatography of recombinant proteins and cytosolic or mitochondrial cellular fractions, respectively. The recombinant protein (500 ul) was injected in a Superdex 75 10/300 GL equilibrated with a gel filtration buffer (20 mM HEPES, 150 mM KCl, pH 7.2). Cytosolic cellular fraction (2.5 mg) or mitochondrial cellular fraction (1 mg) were applied to a Superdex 200 10/300 GL equilibrated with a mitochondrial isolation buffer IBc (10 mM Tris HCl, 1 mM EGTA, 200 mM sucrose, pH 7.5). Fractions of 500 μl were collected, 6.25 μl of 4X LDS/DTT loading buffer added to 18.75 μl of each fraction and analyzed by SDS-PAGE and immunoblotting using the BAX antibody. Gel filtration molecular weight markers (ovalbumin and carbonic anhydrase) were injected to the columns to obtain a standard curve for the estimation of the molecular weight of the proteins. All gel filtration operations were run at 4 °C. Elution fractions for BAX species: BAX monomer (C7, C8, C9), BAX dimer (C3, C4, C5) and BAX oligomer (C2, C1, B1, B2) were determined based on the estimated molecular weight. % BAX monomer = (C6/2 + C7 + C8 + C9) / total elution fractions *100 and % BAX dimer = (C3 + C4 + C5 + C6/2) / total elution fractions *100.

### Blue native PAGE
Cells ~85% confluent were harvested, resuspended in the permeabilization buffer (20 mM HEPES/KOH, pH 7.5; 250 mM sucrose; 50 mM KCl; 2.5 mM MgCl₂) supplemented with 0.025% digitonin and Halt protease inhibitor cocktail (Thermofisher Cat. 1861279) and incubated on ice for 10 min. The cell lysates were centrifuged at 13,000 $g$ for 5 min at 4 °C to pellet the membranes and the supernatant was collected as the cytosolic fractions. 25 μg of the cytosolic fraction was incubated with vehicle (1% DMSO), BIM SAHB or indicated small molecules in a Biorad C1000 Touch Thermal Cycler at 30 °C for 1 h, and on ice for BIM SAHB samples. The samples were then mixed with Invitrogen NativePAGE 5% G-250 Sample Additive (Coomassie blue G-250) to a final concentration of 0.25%. The anionic dye Coomassie blue G-250 binds to proteins and confers them a negative charge promoting their migration to the anode during the electrophoresis. The samples were separated on a Novex NativePAGE 4–16% Bis-Tris Protein Gels

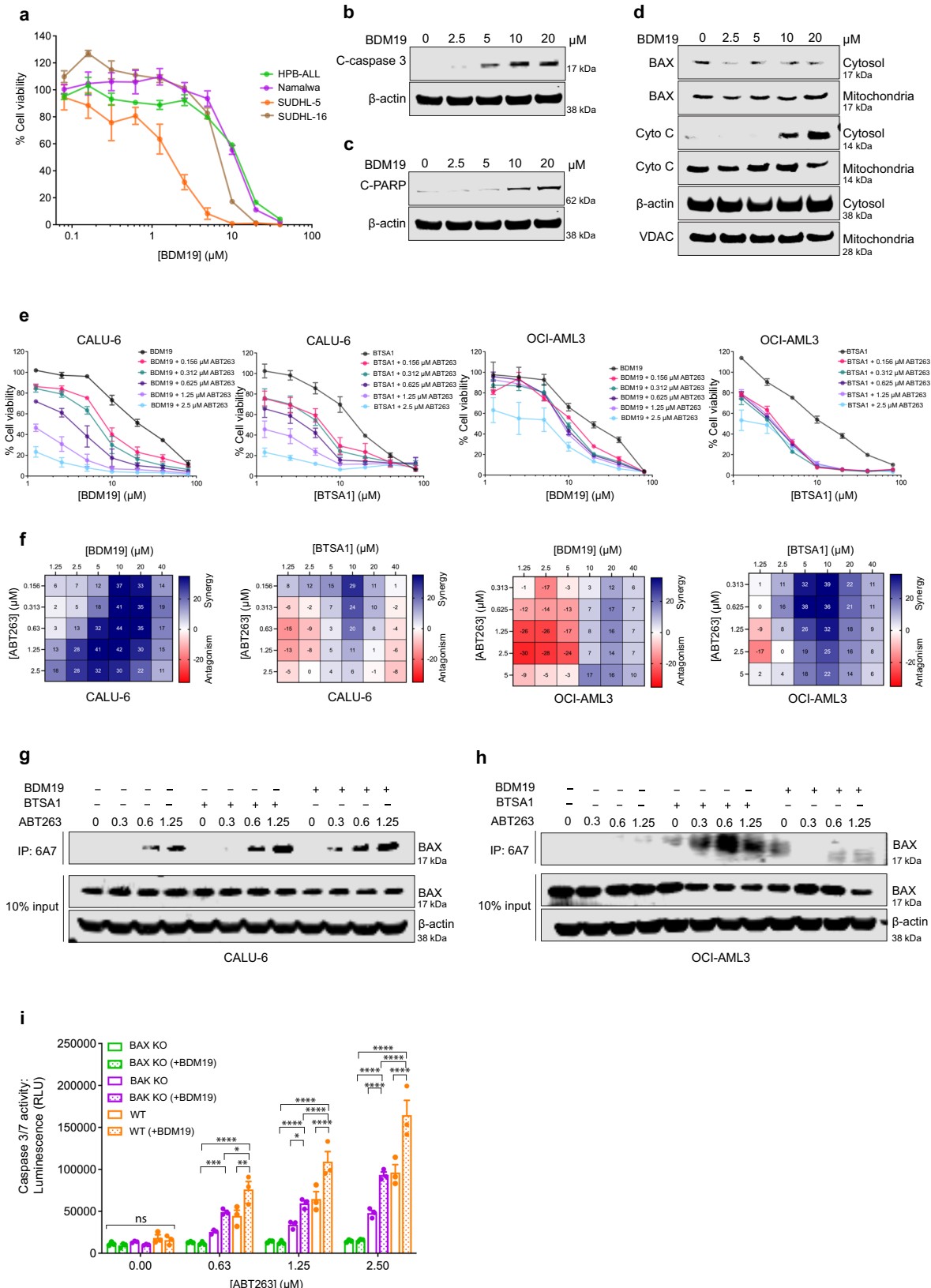

(Invitrogen Cat. BN1004BOX) at 4 °C. The gels were run at 40 V for 30 min and then at 100 V for 30 min with dark cathode buffer (1X NativePAGE Running Buffer and 1X NativePAGE cathode buffer additive [Invitrogen]). Dark cathode buffer was replaced with light cathode buffer (1X NativePAGE Running Buffer and 0.1X NativePAGE cathode buffer additive [Invitrogen]), and the gel was run at 100 V for 30 min

and subsequently at 200 V for 84 min or until the dye front ran off the gel. The cathode buffer additive in the dark and light cathode buffer also contains Coomassie blue G-250 to maintain the negative charge of proteins throughout the BN PAGE run. After electrophoresis was complete, the gels were equilibrated in the transfer buffer (1X NuPage transfer buffer with 20% methanol) supplemented with 0.037% SDS for

**Fig. 7 | BDM19 potentiates BAX activation and apoptosis. a** Cell viability (cell-titer glow assay) of leukemia and lymphoma cell lines (HPB-ALL, Namalwa, SUDHL-5 and SUDHL-16) treated with a titration of BDM19 at 72 hrs. Data are mean ± SEM from $n = 3$ independent experiments. **b** Caspase-3 cleavage in SUDHL-5 cells treated with DMSO or BDM19 for 6 hrs. Whole cell lysates were analyzed by western blot with anti-cleaved caspase-3. **c** PARP-cleavage in SUDHL-5 cells treated with DMSO or BDM19 for 6 hrs. Whole cell lysates were analyzed by western blot with anti-cleaved PARP. **d** BAX translocation to the mitochondria and cytochrome *c* release in the cytosol. SUDHL-5 cells were treated with DMSO or BDM19 for 6 hrs. The cytosol and mitochondria fractions were separated and analyzed by western blot with indicated antibodies. **b**–**d** Blots are representative of at least $n = 2$ independent experiments. **e** Cell viability (cell-titer glow assay) dose-response curves of BDM19 or BTSA1 in the presence of various doses of ABT263 in a cell line with cytosolic BAX dimer, CALU-6, and a cell line with cytosolic BAX monomer, OCI-AML3 at 72 hrs.

Data are mean ± SEM from $n = 3$ independent experiments. **f** Bliss synergy score heatmap of combination studies of BDM19 or BTSA1 and ABT263 combination in (**e**), Data are from $n = 3$ independent experiments. **g**, **h** Active BAX immunoprecipitation with 6A7 antibody of a cell line with cytosolic BAX dimer, CALU-6 (**g**), and a cell line with BAX monomer, OCI-AML3 (**h**). CALU-6 and OCI-AML3 were treated with a combination of titration of ABT263 and a fixed dose of 2.5 μM BTSA1 or 5 μM BDM19 for 2 hrs. Blots are representative of $n = 3$ independent experiments with similar results. **i** Caspase 3/7 activity in CALU-6 WT, CALU-6 CRISPR/Cas9 BAX KO and CALU-6 CRISPR/Cas9 BAK KO cell lines after 8 hrs treatment with ABT263 alone and in combination with a fixed sensitizing concentration of BDM19 (loss of viability <10%). Data are mean ± SEM of three technical replicates from $n = 3$ independent experiments. Statistics were obtained using two-way ANOVA: ns, $p \geq 0.05$; *$p < 0.05$; **$p < 0.01$; ***$p < 0.001$; ****$p < 0.0001$. Source data are provided.

---

10 min to ensure protein denaturation and a negative charge on proteins. The gels were transferred to the Polyvinylidene fluoride (PVDF) membrane at 34 V for 2 h in the transfer buffer or for 7 minutes using the Biorad trans-turbo system. The membranes were incubated with 8% acetic acid for 15 min and washed with $H_2O$ for 5 min. The membranes were then dried at 37 °C for 20 min, rehydrated in 100% methanol and washed in $H_2O$. Next, the membranes were stained with Ponceau and incubated with a N-terminal BAX antibody (Cell Signaling Cat. 2772). The NativeMark™ Unstained Protein Standard (Invitrogen Cat. LC0725) was used to estimate the molecular weight of BAX species.

### Western blotting and quantification

Cells were lysed with 1% Triton lysis buffer (50 mM Tris HCl, 150 mM NaCl, 5 mM $MgCl_2$, 1 mM EGTA, 10% glycerol, 1% Triton, pH 7.4) supplemented with Halt protease inhibitor cocktail (Thermofisher Cat. 1861279). Whole-cell lysates were electrophoretically separated on 4–12% NuPage gels (Life Technologies), transferred to mobilon-FL PVDF membranes (Millipore) and subjected to immunoblotting. For visualization of proteins with chemiluminescence or Odyssey Infrared Imaging System (LI-COR Biosciences), membranes were blocked in PBS containing 5% milk powder for 1 h at room temperature. Primary antibodies were incubated overnight at 4 °C. After washing, membranes were incubated with HRP anti-rabbit/mouse secondary antibody (1:5,000 dilution) for chemiluminescence or an IRdye800-conjugated goat anti-rabbit IgG (1: 10, 000) / IRDye800- conjugated goat anti-mouse IgG (1:10,000 dilution)/IRDye680RD-conjugated goat anti-Rabbit IgG (1:10,000 dilution) secondary antibodies (LI-COR Biosciences) for the Odyssey Infrared Imaging System. Protein levels were quantified by densitometric analysis (chemiluminescence) or fluorescence intensity (Odyssey Infrared Imaging System) using the Image Studio 3.1 software. Antibodies were used to detect the following proteins on membrane: BAX (Cell Signaling Cat. 2772), BAK (Cell Signaling Cat. 12105), BCL-XL (Cell Signaling Cat. 2764), MCL-1 (Cell Signaling Cat. 5453), BCL-2 (Cell Signaling Cat. 4223), BIM (Cell Signaling Cat. 2933), BID (Santa Cruz Biotechnology Cat. 11423), Cleaved Caspase-3 (Cell Signaling Cat. 9664), Cleaved PARP (Cell Signaling Cat. 5625), COX-IV (Cell Signaling Cat. 4850), β-Actin (Sigma Cat. A1978), β-Tubulin (Cell Signaling Cat. 2146), and VDAC (Abcam Cat. 15895).

### Cellular BAX translocation and oligomerization assay

Cells ~85% confluent were treated with ABT263 at indicated concentrations or vehicle (1% DMSO) for 6 hrs. Cells were then washed with PBS and harvested. The cytosolic and mitochondrial cellular fractions were isolated using a Dounce homogenizer in a mitochondria isolation buffer (IBc) (10 mM Tris HCl, 1 mM EGTA, 200 mM sucrose, pH 7.50) supplemented with Halt protease inhibitor cocktail (Thermofisher Cat. 1861279). Cytosolic cellular fraction (2.5 mg) or mitochondrial cellular fraction (1 mg) were applied to a Superdex 200

10/300 GL equilibrated with the mitochondrial isolation buffer IBc. Fractions of 500 μl were collected, 6.25 μl of 4X LDS/DTT loading buffer added to 18.75 μl of each fraction and analyzed by SDS-PAGE on 4–12% NuPage gels (Life Technologies) and immunoblotting using the BAX antibody (Cell signaling Cat. 2772) overnight at 4 °C. After washing, membranes were incubated with HRP anti-rabbit/mouse secondary antibody (1:5,000 dilution) and visualized using chemiluminescence.

### BAX conformational change assay

Cells were harvested and lysed with a 1% CHAPS lysis buffer (150 mM NaCl, 10 mM HEPES, pH 7.4, 1% CHAPS) supplemented with Halt protease inhibitor cocktail (Thermofisher Cat. 1861279). Whole-cell lysates were subjected to immunoprecipitation followed by immunoblotting against total BAX. Briefly, 500–1000 μg total protein in 600 μl was collected and incubated with 12 μl (50% slurry) of pre-equilibrated protein G agarose beads (Santa Cruz Biotechnology) for 30 min at 4 °C on a rotator. The precleared samples were then incubated with 5 μl of 6A7 antibody (Santa Cruz Biotechnology Cat. 23959) overnight at 4 °C. The next day, 20 μl (50% slurry) of pre-equilibrated G agarose beads (Santa Cruz Biotechnology) were added to the samples and incubated for 2 h at 4 °C. The beads were pelleted, washed with the lysis buffer 3 times, and protein eluted by heating the beads at 90 °C for 15 minutes in 1X LDS/DTT loading buffer. The immunoprecipitates were subjected to electrophoresis on 4–12% NuPage gels (Life Technologies). For visualization of proteins using Odyssey Infrared Imaging System (LI-COR Biosciences), membranes were blocked in PBS containing 5% milk powder for 1 hour at room temperature. Primary antibodies were incubated overnight at 4 °C. After washing, membranes were incubated with an IRdye800-conjugated goat anti-rabbit IgG (1:10, 000) or IRDye800- conjugated goat anti-mouse IgG (1:20,000 dilution) secondary antibodies (LI-COR Biosciences). Antibodies were used to detect the following proteins on membrane: BAX (Cell Signaling Cat. 2772) and β-Actin (Sigma Cat. A1978) or β-tubulin (Cell Signaling Cat. 2146).

### Viral transduction

HCT116 BAX KO cells were reconstituted with human BAX WT or BAX mutants (BAX P168G and BAX E75K) using retroviral transduction with BAX-IRES-GFP, followed by Becton Dickinson FACS Aria single-cell sorting based on GFP levels. Cells with similar GFP levels were combined for each cell line (WT or mutants). Comparable BAX protein levels of the WT and mutants were confirmed by western blot analysis. Bax and Bak DKO MEFs expressing BAX WT, R134E, R145E were generated using the same protocol. DKO MEFs expressing K21E were generated previously[15].

### Cell viability assay

Cells ($2 \times 10^3$ cells/well) were seeded in 384-well white plates overnight and incubated with serial dilutions of Navitoclax (ABT263), BTSA1,

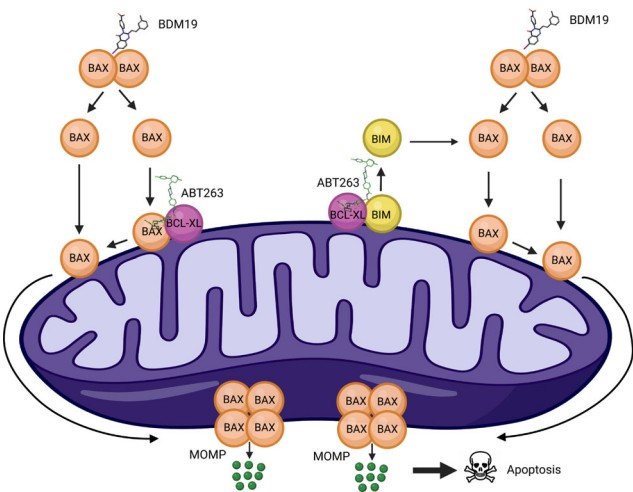

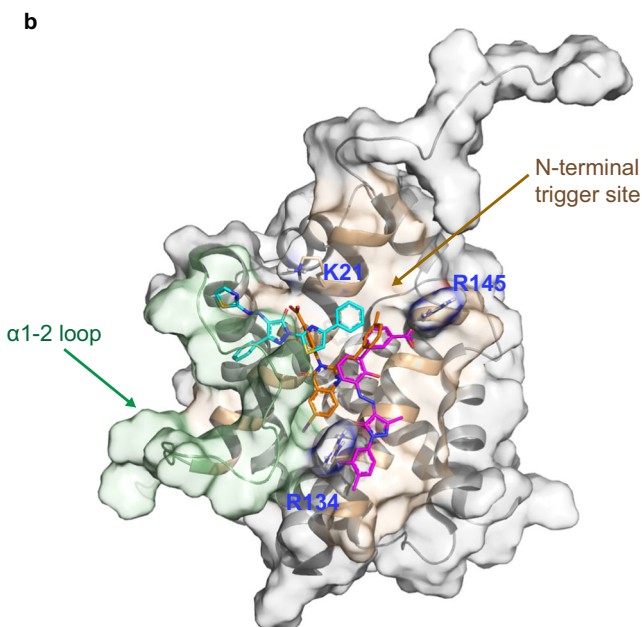

**Fig. 8 | BDM19 potentiates BAX-mediated apoptosis through distinct binding at the N-terminal trigger site of BAX compared to other BAX small molecules. a** Schematic showing the mechanisms of BAX-induced apoptosis by BDM19 and its combination with ABT-263. This figure was created with BioRender.com. **b** Binding mode of BDM19 (orange) on BAX structure (PDB: 1F16) in comparison to small molecules that bind at the N-terminal trigger site BTSA1 (teal) and eltrombopag (violet). The N-terminal trigger site is highlighted in khaki (helix α1 and α6) and green (α1-2 loop) for clarity. The positive charge residues at the N-terminal trigger site K21, R134 and R145 that interact with BDM19, and in part with BTSA1 (K21) and eltrombopag (R134 and R145), are shown in sticks for clarity.

BDM19 or vehicle (1% DMSO) using a TECAN D300e Digital Dispenser in 25 ul. In the case of BDM19 and BTSA1, no FBS media was used, followed by the addition of 10% FBS after 2.5 hrs for comparison with established conditions for BTSA1 due to its high binding to FBS[39]. Cell viability was assessed at 24 hrs or 72 h as indicated, by addition of CellTiter-Glo Assay reagents according to the manufacturer's protocol (Promega), and luminescence measured using a F200 PRO microplate reader (TECAN). For the Navitoclax (ABT263) and BDM19 or BTSA1 combination experiments, cells were seeded as described above and

co-treated with Navitoclax (ABT263) and BDM19 or BTSA1 at the indicated doses. Viability assays were performed in at least triplicate and the data normalized to 1% vehicle-treated control wells. IC50 values were determined by nonlinear regression analysis using GraphPad Prism software 9 (GraphPad). The BLISS calculation was determined using the Combenefit program as previously described[50].

### Caspase 3/7 activation assay

Cells ($2.5 \times 10^3$ cells/well) were seeded in 384-well white plates overnight and incubated with serial dilutions of Navitoclax (ABT263), A-1331852, ABT199, doxorubicin, BDM19 or vehicle (1% DMSO) using a TECAN D300e Digital Dispenser, in 25 µl of media with 10% FBS. For BDM19, the compound was dispensed in media without FBS and 10% FBS was added after 2 hrs. For the BDM19 and doxorubicin combination, cells were seeded as described above and co-treated with doxorubicin and BDM19 in media without FBS and 10% FBS was added after 2 h. Caspase 3/7 activity was assessed at 2, 6, 8 or 36 hrs as indicated, by the addition of Caspase 3/7 Assay reagents according to the manufacturer's protocol (Promega), and luminescence measured using a F200 PRO microplate reader (TECAN).

### Annexin-V staining assay

Cells ($2.5 \times 10^3$ cells/well) were seeded in 384-well white plates overnight and incubated with serial dilutions of Navitoclax (ABT263) or vehicle (1% DMSO) using a TECAN D300e Digital Dispenser, in 25 µl of media with 10% FBS. Annexin V staining was assayed at 2 h, by the addition of RealTime-Glo™ Annexin V Apoptosis and Necrosis Assay reagents according to the manufacturer's protocol (Promega), and luminescence measured using a F200 PRO microplate reader (TECAN). Relative % annexin V staining was obtained by normalization to the luminescence of the 1% vehicle-treated control wells.

### Depolarization assay/TMRE staining

Cells ($1 \times 10^5$ cells/well) were seeded in black 96-well plates overnight and incubated with a serial dilution of Navitoclax (ABT263) or the vehicle (1% DMSO) using the Tecan D300E digital dispenser in 100 ul of media with 10% FBS. After 6.5 h, 100ul of TMRE in PBS was added to each well to a final concentration of 250 nM, incubated for 20 min at 37 °C and washed three times with 200 µl PBS. 100 µl of PBS was added to each well and fluorescence read by a M1000 microplate reader (TECAN) at 540 nm excitation and 595 nm emission. Relative TMRE fluorescence was obtained by normalization to the 1% vehicle-treated control wells.

### Cytochrome C release assay

SUDHL-5 cells and HCT116 BAX KO reconstituted with BAX WT (cytosolic dimer) or mutants P168G (cytosolic dimer) or E75K (cytosolic monomer) were treated with BDM19 or ABT-263 respectively at indicated concentrations or vehicle (1% DMSO). In the case of BDM19, media without FBS was used, followed by the addition of 10% FBS after 2 hrs for comparison with established conditions for BTSA1 due to its high binding to FBS[39]. After 6 hrs of treatment for BDM19 in SUDHL5 and 3 h for ABT263 in HCT116 BAX mutants, cells were washed with PBS and harvested. The cytosolic and mitochondrial cellular fractions were isolated using centrifugation in a digitonin lysis buffer (10 mM KCl, 5 mM MgCl$_2$, 1 mM EGTA, 1 mM EDTA, 250 mM sucrose, 20 mM Hepes, 0.025% digitonin, pH 7.2) supplemented with Halt protease inhibitor cocktail (Thermofisher Cat. 1861279). Cells were incubated with the lysis buffer for 30 min on ice and centrifuged at 15,000 g for 10 min at 4 °C. Then, the cytosolic fraction (supernatant) collected. The pelleted membranes were dissolved in 1% triton X-100 for 1 hr on ice, centrifuged at maximum speed for 15 minutes at 4 °C and solubilized mitochondria fraction collected. Samples were analyzed by SDS-PAGE on 4–12% NuPage gels (Life Technologies) and immunoblotting using primary antibodies overnight at 4 °C. After washing, membranes

were incubated with an IRdye 800RD-conjugated goat anti-rabbit IgG (1:10,000) or IRDye 680RD- conjugated goat anti-mouse IgG (1:10,000 dilution) secondary antibodies (LI-COR Biosciences). Antibodies were used to detect the following proteins on membrane: BAX (Cell Signaling Cat. 2772), cytochrome c (BD Pharmingen Cat. 556433), β-Tubulin (Cell Signaling Cat. 2146), and VDAC (Abcam Cat. 15895).

## Mitochondrial priming

BIM BH3 (0.1 µM); Puma2A peptide (final concentration of 10 µM); alamethicin (final concentration of 25 µM) and CCCP (final concentration of 10 µM) were added to JC1-MEB staining solution (150 mM mannitol, 10 mM HEPES-KOH, 50 mM KCl, 0.02 mM EGTA, 0.02 mM EDTA, 0.1% BSA, 5 mM succinate, pH 7.5) in a black 384-well plate. Single cell suspensions of various cell lines were prepared in JC-1-MEB buffer as described previously[51]. Cells were kept at room temperature for 10 min to allow for cell permeabilization and dye equilibration. After adding the cells to the 384-well plate ($2.0 \times 10^4$ cells/well to $4.0 \times 10^4$ cells/well), fluorescence was measured at 590 nm emission 545 nM excitation using the M1000 microplate reader (TECAN) at 30 °C every 15 min for a total of 3 h. The percentage of depolarization was calculated by normalization to the area under the curve (AUC) of solvent-only control DMSO (0% depolarization) and the positive control CCCP (100% depolarization).

## In silico small molecule pharmacophore-based screening

eMolecules (www.emolecules.com) library of purchasable compounds was converted to 3D structures using LIGPREP (LigPrep, Schrödinger Release 2018, Schrödinger, LLC) and EPIK (Epik, Schrödinger Release 2018, Schrödinger, LLC) sampling different ionization states at pH $7.0 \pm 2.0$, stereochemistry and tautomeric forms resulting in an in silico library of approximately 14 million screening compounds. Conformation analysis of ligands was calculated using the OPLS3 force field. Phase (Phase, Schrödinger Release 2018, Schrödinger, LLC) module was used to perform 3D pharmacophore screens. The interface of the autoinhibited BAX dimer crystal structure (PDB 4S0O) was analyzed for residue interactions and the coordinates of the several residues: Q28, L45, A46, L47, D48, R109, Y164, T172 and F176 were used to assign pharmacophore points in 3D coordinates. The pharmacophore points included hydrophobic groups to mimic the sidechains of Q28, L45, L47 and T172 residues, aromatic group to mimic the sidechain of F176, positively and negatively charged groups to mimic the side of R109 and D48 and hydrogen bond donor and acceptor groups to mimic the mainchain of A46 and sidechain of Y164. Pharmacophore-based screens require compounds to satisfy 4-5 pharmacophore points using different 3 hydrophobic groups, one aromatic group and one hydrogen bonding or charged group. Based on PhaseScore, the top 1000 ranked compound hits were selected for further visual analysis and clustered for diversity using dendritic fingerprints in Canvas. Physicochemical and AMDET properties including Lipinski rules, permeability, logP, metabolic liabilities, and hERG inhibition were evaluated using QikProp (QikProp, Schrödinger Release 2018, Schrödinger, LLC). Among highest rank compounds 26 diverse structures were selected for experimental validation. Compounds were checked for potential Pan Assay Interference Compounds (PAINS) and BDM19 was confirmed to have not been reported as a hit in previous screens in the PubChem database.

## Recombinant BAX protein production

Human full-length (aa1–192) wild-type BAX was cloned in pTYB1 vector (New England BioLabs) between the NdeI and SapI restriction sites. Mutations were generated using the QuickChange Lightning site directed mutagenesis kit (Agilent). Recombinant proteins were expressed in BL21 (DE3) CodonPlus (DE3)-RIPL, grown in Luria Broth media and induced with 1 mM isopropyl β-d-1-thiogalactopyranoside (IPTG). The bacterial pellet was resuspended in lysis buffer (20 mM

Tris–HCl, 250 mM NaCl, pH 7.2 and Roche complete EDTA free protease inhibitor cocktail), lysed by high pressure homogenization, and clarified by ultracentrifugation at 19 000 rpm for 45 min. The supernatant was applied to 5 ml of pre-equilibrated chitin beads (New England BioLabs) in a gravity-flow column and washed with 3 column volumes of lysis buffer. BAX was cleaved by overnight incubation using 50 mM DTT in lysis buffer. Cleaved BAX was eluted with lysis buffer, concentrated with a Centricon spin concentrator (Millipore) and purified by gel filtration using a using a Superdex 75 10/300 GL column (Cytiva), pre-equilibrated with gel filtration buffer (20 mM HEPES, 150 mM KCl, pH 7.2) at 4 °C. Fractions containing BAX monomer were combined and concentrated using a 10-kDa cut-off Centricon spin concentrator (Millipore) for prompt use in biochemical and structural studies. For Microscale Thermophoresis, BAX S4C C62S C126S (BAX 4 C) or BAX 4 C mutants (BAX 4 C K21E, BAX 4 C R134E, BAX 4 C R145E, BAX 4C R134E R145E and BAX Q28A) was purified as described above with additional 5% glycerol in the lysis buffer (20 mM Tris–HCl, 250 mM NaCl, 5% glycerol, pH 7.2 and Roche complete EDTA free protease inhibitor cocktail) along with 5% glycerol and 0.5 mM TCEP in the gel filtration buffer (20 mM HEPES, 150 mM KCl, 5% glycerol, 0.5 mM TCEP, pH 7.2). Fractions containing BAX monomer were combined, concentrated to 70 µM using a 10-kDa cut-off Centricon spin concentrator (Millipore) and frozen at −80 °C in 50 µl aliquot for further use.

## Fluorescence polarization binding assays

Fluorescence polarization assays (FPA) were performed as previously described[12,38]. Firstly, direct binding curves were generated by incubating FITC-BIM SAHB (25 nM) with serial dilutions of full-length BAX. Fluorescence polarization was measured every 10 min for 60 minutes on a F200 PRO microplate reader (TECAN). Reported curves are the 10-minute time-point. For the competition assay in the screen, 26 compounds at 10 µM were combined with recombinant BAX at EC75 concentration as determined by the direct binding assay (BAX: 125 nM) followed by the addition of FITC-BIM SAHB (25 nM). For titration assays of the top hits, a serial dilution of BDM19 or analogues was combined with recombinant BAX at EC75 concentration as determined by the direct binding assay (BAX: 125 nM) followed by the addition of FITC-BIM SAHB (25 nM). $EC_{50}$ values were calculated by nonlinear regression analysis four-parameter agonist versus response with restraints for 100% top and 0% bottom calculated by the mP of saturated BAX + FITC-BIM-SAHB and FITC-BIM-SAHB alone, respectively. Data were analyzed and graphed using GraphPad Prism 9 software.

## Microscale thermophoresis

Recombinant BAX S4C C62S C126S (BAX 4 C)[41,42], previously established for the assessment of BAX binding compounds with Microscale thermophoresis (MST), or BAX 4 C mutants (BAX 4 C K21E, BAX 4 C R134E, BAX 4C R145E, and BAX Q28A) were labeled at cysteine using the Monolith Protein Labeling Kit Red Maleimide 2nd generation (NanoTemper Technologies) according to the instructions of the manufacturer. Specifically, 35 µM BAX was incubated with around 2 equivalents of dye in MST buffer (100 mM potassium monophosphate, 150 mM NaCl, pH 7.4) in the dark at room temperature (22–25 °C) for 30 min. Unreacted dye was quenched using 7 mM DTT and removed using the superdex 75 10/300 GL column (Cytiva). To determine the KD of BAX to BDM19 or analogs BDM19.1, BDM19.2, BDM19.3 and BDM19.4, 50 nM labeled BAX was incubated with increasing concentrations of BDM19 or analogs in MST buffer supplemented with 0.25% CHAPS. For experiments using activated BAX, 50 nM labeled BAX was incubated with increasing concentrations of BDM19 in the MST buffer supplemented with 1% NP40 and incubated for 10 min. Samples were loaded into standard glass capillaries (Monolith NT.155 Capillaries) and analyzed by MST using a Monolith NT.115 Blue/Red, LED power and IR laser power of 40%. Samples showed no aggregation

according to post-run analysis using the Monolith data collection software (Nanotemper). Fraction bound and error were generated by NanoTemper software (MO.Affinity Analysis) and KD values were determined using GraphPad Prism 9 and nonlinear fit of one-site specific binding.

## Cellular BAX engagement assay

DKO MEFs expressing BAX WT or mutants K21E, R134E and R145E cells were seeded in a 10 cm dish until approximately 85% confluent. The cells were washed and harvested in PBS. $7 \times 10^6$ Cells were incubated in 60 μM of BDM19 dissolved in PBS for 1 hour at room temperature on a rotator. 50 μl of the sample was transferred to PCR-tubes and heated in a Biorad C1000 Touch Thermal Cycler for 3 minutes using a temperature gradient (50, 52.1, 55.4, 59.4, 64.9, 69.2, 72.1, and 74 °C). Cells remaining at room temperature (25 °C) served as a control. Cells were lysed by three cycles of freeze thaw using liquid nitrogen and centrifuged at $2 \times 10^4$ g for 15 min. The supernatants were collected, resolved by SDS-PAGE on 4–12% NuPage gels in a 1X LDS/DTT loading buffer (Life Technologies) and immunoblotted using an N-terminal BAX antibody (Cell Signaling Cat. 2772). Proteins were visualized using the Odyssey Infrared Imaging System (LI-COR Biosciences), fluorescence intensity quantified using the Image Studio 3.1 software, and normalized to 25 °C (100%) and 74 °C (0%).

## NMR samples and spectroscopy

For NMR studies, uniformly $^{15}$N-labeled BAX protein was produced by growing the bacteria in minimal medium and $^{15}$NH$_4$Cl as previously described[52]. Following similar purification of BAX as with unlabeled protein, $^{15}$N-labeled BAX protein samples (yield ~0.25 mg protein per 1 L of M9 media) were buffer exchanged to 50 mM potassium phosphate, 50 mM NaCl solution at pH 6.0 in 10% D$_2$O. $^1$H-$^{15}$N-HSQC experiments were performed using an independent sample for each experimental measurement in a 5-mm Shigemi; all samples were DMSO matched with 2% d$_6$-DMSO. $^1$H-$^{15}$N-HSQC spectra were recorded on $^{15}$N-labeled BAX at 50 μM in the presence and absence of 100 μM and 200 μM of BDM19. NMR spectra were acquired at 25 °C (298.15 K) on a Bruker 600 MHz spectrometer equipped with a cryoprobe, processed using TopSpin 3.6.2 and analyzed using CcpNmr Analysis 2.5.2. BAX chemical shifts assignments were applied as previously generated for BAX[17,18] and deposited (BMRB Entry 4632) and buffer matching and acquisition temperature were considered to minimize errors. Optimized parameters during data collection are spectral Width= 14.0289ppm (F2) and 29.0001ppm (F1); acquisition time= 0.1216512 sec (F2) and 0.0283548 sec (F1); number of scans= 32; number of dummy scans= 16. The weighted average chemical shift perturbation (CSP) was calculated as $\sqrt{[(\Delta\delta^1H)^2 + (\Delta\delta^{15}N/5)^2]/2}$ in p.p.m and the significance threshold for backbone amide chemical shift changes was calculated based on the average chemical shift across all residues plus 0.5 or 1 s.d.[41]. The absence of a bar indicates no chemical shift difference, the presence of a proline, or a residue that is overlapped or missing and therefore not used in the analysis. Mapping of chemical shifts onto the BAX structure was performed with PyMOL (Schrodinger, LLC, 2022). The software was made available through the SBGrid collaborative network[53].

## Molecular docking and molecular dynamics simulations

NMR-guided docking of BDM19 into the NMR structure of BAX (PDB: 1F16) was performed using induced-fit docking (IFD, Schrodinger, LLC, 2020) with extra precision (XP) and a binding site at the mid-point of residues K21, R134, and R145. BDM19 or BTSA1 was converted to a 3D all atom structure using LIGPREP (Schrodinger, LLC, 2020) and assigned partial charges with EPIK (Schrodinger, LLC, 2020). Poses generated indicated ionic interaction between the carboxylate of BDM19 and a basic residue of BAX, K21 as well as the carbonyl of BDM19 with a polar residue of BAX, Q28. Mutagenesis was used to

elucidate the pose of BDM19 on the trigger site of BAX. The top-scoring pose which was consistent also with the MST binding data using BAX mutants and with NMR CSP data was chosen. For BTSA1, the top pose was consistent with the published binding pose[35]. Each pose was subjected to three independent 1μs molecular dynamics (MD) simulations starting with different seeds using DESMOND (DESMOND, version 3, Schrodinger, LLC, 2020-21). Three independent 1 μs MD simulations were also performed with the lowest energy BAX structure from the NMR ensemble (PDB 1F16) starting with different seeds. MD runs were performed in a truncated orthorhombic box of size $10 \times 10 \times 10$ Å with a total 366343 Å$^3$ minimized volume with TIP3P water using OPLS4 force field, 300 K, and the constant pressure of 1.0325 bar. The system was neutralized by adding 3 Na$^+$ ions and a salt concentration of 0.15 M of NaCl was included in the simulation box. For BAX alone, the system consisted of 35203 atoms and 10721 water molecules. For the BAX-BDM19 complex, the system consisted of 33650 atoms and 10189 water molecules. For the BAX-BTSA1 complex, the system consisted of 39298 atoms and 12068 water molecules. The Nose–Hoover Chain thermostat and Martyna–Tobias–Klein barostat were used to maintain the temperature and pressure, respectively. Analysis of the trajectory was performed with MAESTRO simulation event analysis tools (Schrodinger, LLC, 2020-21). The ΔRMSF for each residue was calculated as ΔRMSF = ((RMSF BDM19 −RMSFApo)/RMSFApo), where RMSF BDM19 was the RMSF of an individual MD simulation of BDM19 docked into BAX and RMSFapo is the average RMSF of the apo BAX simulation. The area of the opening of canonical site was calculated as area of canonical site = area of triangle K189-V91-T85 + area of triangle K189-M79-T85 with the area of the triangle calculated using Heron's formula. Interatomic distances, energies, radius of gyration, ΔRMSF and RMSD data obtained from MD analysis were plotted using GraphPad Prism 9. PyMOL (Schrodinger, LLC, 2020-21) was used for preparing the highlighted poses.

## Statistical analysis

Plots and statistical tests were generated in GraphPad Prism 9.0. Data are presented as means ± SD except where noted. Statistical significance for pair-wise comparison of groups was determined by two-way ANOVA using GraphPad Prism 9 software (GraphPad Inc) unless otherwise indicated. $P$ values correspond to symbols as follows: ****$p < 0.0001$; ***$p < 0.001$; **$p < 0.01$; *$p < 0.05$; ns $P > 0.05$. Additional statistical details are provided in figure legends and methods details.

## Reporting summary

Further information on research design is available in the Nature Portfolio Reporting Summary linked to this article.

## Data availability

Data generated or analyzed during this study are included in this published article and its supplementary information files and are available from the corresponding author on a reasonable request. Molecular dynamics simulation input files, starting and final structures for each system are provided in Open Science Framework depository under accession number https://doi.org/10.17605/OSF.IO/48WKE[54]. The following publicly available data sets were used in the production of this manuscript: PDB 4S0O https://doi.org/10.2210/pdb4S0O/pdb, PDB 1F16 https://doi.org/10.2210/pdb1f16/pdb. Source data are provided with this paper.

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

## Acknowledgements

The authors thank all members of the Gavathiotis laboratory for their advice during this project. Studies were supported by the Irma T. Hirschl Trust Career Award (E.G.) and National Institutes of Health grants R01CA223243 (E.G.), PR191593P1 (E.G.), R01CA178394 (E.G.), P01AG031782 (E.G.) and R01 CA223231 (E.H.C.). Support of various facilities was provided by P30CA013330, P30CA008748 and S10OD01630 grant for the NMR resources. Images in Figs. 1a, 8a. and Supplementary Fig. 3c were created with BioRender.com.

## Author contributions

N.G. performed biochemical, molecular modeling and cell-based studies. B.A. performed pharmacophore screening and assisted in biochemical studies. O.W.M., V.K.M.V. performed NMR-based studies. E.C. provided MEF cell lines. E.G. conceived and designed the study, wrote the manuscript with N.G., which was edited by all authors.

## Competing interests

E.G. has received compensation for consulting and serving on scientific advisory boards, and has equity ownership from BAKX Therapeutics, Life Biosciences, Stelexis Therapeutics; and has consulted for Boehringer Ingelheim and Guidepoint. All E.G. consulting activities are outside of the submitted work. The other authors declare no competing interests.
