## [Peer Review File · Nature Communications]

Chemical modulation of cytosolic BAX homodimer potentiates BAX activation and apoptosisREVIEWER COMMENTS

Reviewer #1 (Remarks to the Author):

This manuscript described a small-molecule modulator, BDM19 that binds BAX, activates cytosolic dimers and prompts cells to BAX-mediated apoptosis alone or in combination with BCL-2/BCL-XL inhibitor Navitoclax. By using BDM19, the authors made many efforts to try to underscore the role of the cytosolic inactive BAX dimer in resistance to apoptosis of cancer cells and discussed the structure basis of Bax dimer activation. However, there are some concerns needed to be addressed. I recommend a major revision.

1. Since BTSA1 only activates Bax monomer rather than dimer, a solid structural basis of Bax dimer activation could have been achieved by comparing with BMD19. Do not only suggest that R134 and R145 are the hot spots because they are the additional residues bound by BMD19, but test the mutants. The identified new hot spot is the most valuable contribution in this study to the field of Bcl-2 family. Alternatively, based on the binding mode predicted by the authors, 3-Toluene and quinazoline of BDM19 could form cationic - π interaction with amino acid residues R145 and R134. Therefore, replacing aromatic group with molecules that cannot form a connection - π interaction with R145 and R234, such as 3-methylepoxide or tetrahydroquinazoline, could verify whether R145 and R134 are the key amino acid residues for Bax dimer activation.
2. The Bax activation by BDM19 in Figure 7g is not concentration-dependent. It challenges BDM19's Bax activation.
3. NMR, MD and FP demonstrated that the binding sites of BDM19, BTSA1 and Bim are highly overlapped. All these compounds bind in the groove between $\alpha 1$ and $\alpha 6$, and they all interact with residues K21, M20, M137 and L141. As such, it is not proper to claim that BMD19 has a distinct binding site with other Bax activators.
4. Since MD result suggested that BDM19 induces Bax conformation change and activation similar as BTSA1, why BMD19 cannot induce Bax monomer and apoptosis in OMI-AML3 cell lines? Why the Bax monomer activation by BMD19 is weaker than BTSA1?
5. The author run 1000 ns in MD but RMSD still indicates that the system did not reach the equilibrium phase. The reliability of MD is weakened.
6. Two cell lines (SUDHL-16 and Namalwa) used for cell experiments in Figure 7 are not in Figure 1, so the level of Bax dimer and monomer there is unclear. This reviewer suggests the author use the cell lines in Figure 1.
7. It is strange Eltrombopag could improve the dynamic of Bax conformation like BMD19 as the author claimed at Page7, Line402. It is established that the Bax inhibitor Eltrombopag stabilizes the inactive conformation of Bax.
8. Please provide the NMR, HPLC and MS data and spectrum of the final compound BMD19.

9. Many editing mistakes need to be improved.

Reviewer #2 (Remarks to the Author):

The authors present a wonderful story about the role of cytosolic inactive Bax monomers and dimers in avoiding cancer by tumor cells and an elegant way of activation those species by small effector ligands to become active and overcome apoptotic hindrance. In this nice piece of work they also highlight many aspects of the regulation of those species and of activated Bax in various cells lines and how those species are linked to each other and their apoptotic activity upon translocation to the mitochondria for membrane perforation and blocking anti-apoptotic proteins like Bcl-2 e.g. to protect cells against cell death. In their well designed and executed study they use a wide range of powerful methods starting from cell assays to in silico drug screening, NMR and other binding assays and even MD simulations. And indeed, the results and how they are presented is very impressive.

From my point of perspective there are only a few minor points the authors show consider.

1. They authors used a range of cancer cell lines. I wonder how they have selected them and if it would have been possible to use a non-cancer cell line reflecting normal cells for comparison e.g. Many cells who undergo apoptosis are not cancer cells but normal cells e. g. like under embryogenesis.

2. They authors used Navitoclax which inhibits Bcl-2/Bcl-XL. I wonder if the authors tried also the Bcl-2 specific Venetoclax inhibitor or would Bcl-xl have compensated for that in the authors assays.

3. What was the concentration of the Bax samples used in NMR measurements and what was the yield of the ¹⁵N-labelled Bax by their overexpression systems used?

4. In the introduction the authors mentioned Bax mutations P168A and G67R found in AML patients and interfering with Venetoclax. Since there are also mutation G101V (1) found in Bcl-2 preventing Venetoclax working properly, I wonder if the authors might have a molecular idea about how Bax mutants interfere there. I always thought (quite simplified probably) that Venetoclax binds to Bcl-2 and Bax gets just released to do its work. But this might be too simple thought by me.

5. Line 223 "cytosolic inactive Bax dimer constitutes a road block to Bax activation.."

Do the authors know why apoptotic regulation would need this kind of mechanism: how are the levels of inactive Bax dimers in the cell lines used by the authors. And how are those levels compared to non-cancer cells lines (what feeling do the authors

have about those levels; does not need to have been measured before)?

6. Lines 324-339: very detailed molecular description: could be shortened I believe.

7. Line 505...: What happens to the BDM19 ligand after binding to inactive Bax dimers and converting them into active ones who can trigger apoptosis. Is the ligand losing its affinity upon conversion of Bax and if not is Bax then carry out mitochondrial outer membrane formation with the ligand attached to it (could Bcl-2 bind activated Bax with BDM19 ligand attached):

Ref.

1. R. W. Birkinshaw, J. N. Gong, C. S. Luo, D. Lio, C. A. White, M. A. Anderson, P. Blombery, G. Lessene, I. J. Majewski, R. Thijssen, A. W. Roberts, D. C. S. Huang, P. M. Colman, P. E. Czabotar, Structures of BCL-2 in complex with venetoclax reveal the molecular basis of resistance mutations. *Nat Commun* 10, (2019).

Reviewer #3 (Remarks to the Author):

Gitego et al. report their new research on a pro-apoptotic protein, BAX, extended from the prior work from the Gavathiotis laboratory concerning the inactive form of BAX dimers. They found that BAX can form a dimer that makes it resistant to its activation. Now, they report that this form of BAX is found in the cytosolic fraction of many cancer cell lines and that these cells are more resistant to apoptosis inducers. Anti-cancer therapies targeting BAX are a promising approach, however, the presence of these inactive BAX dimers in the cells would hinder it. In order to probe the mechanism and develop a means to reduce these inhibitory dimers, they screened a compound library and discovered BDM19, which disrupts the inactive BAX dimers. Using structural biological techniques, they show that BDM19 interacts with the BAX trigger site in a different manner from that of BIM BH3 or a BAX activator compound, BTSA1 they previously reported.

The study is carefully conducted and in general the data are of high quality. It would give yet new insights to BAX activation processes as well as new tools to the field of cancer research. In order to make the results more convincing and useful to the field, I would like the following issues to be addressed/clarified:

1. The authors should describe Westernblotting of BN-PAGE in more detail. BN-PAGE takes advantage of Coomassie Blue binding to the proteins to cover the charge residues. Such proteins lose the charge, therefore they are not transferred to the PVDF/nitrocellulose electronically. It is unclear how this hurdle was overcome in the method. Moreover, it is surprising that transferred proteins did not seem to be denatured and yet anti-BAX antibody detected it. These issues should be clarified, as this method would be useful when other investigators wish to detect the inactive BAX dimers who are not necessarily equipped with the more elaborate SEC system.

2. Related to the above, would there be a more convenient way to detect these dimers, using the whole cell lysate without cell fractionation? BAX would be oligomerized if non-ionic detergent such as TX-100 was used to lyse cells, so how about CHAPS or digitonin? Would the use of these detergents be compatible with the detection of the inactive dimers?

3. How would the authors explain the “in-between” size shown in OCI-AML3 cell lysate (Fig. 1b) and HCT116 E75K (1f)? Also, in Fig. 2f, the peak size of BAX in WT cytosol and mitochondria and P168G cytosol appears “in-between” the monomer and dimer, not monomeric, as the authors interpreted (lines 214-219). Please explain.

4. In Fig. 2, various BAX mutants expressing cells were compared in their sensitivity to apoptosis. Do the authors know that BAX P168G, if rendered monomeric, has the same activity as WT? In other words, P168G mutation does not affect the series of changes BAX undergoes during activation, such as conformational change, insertion of α_9 , forming BH3 swapped dimers and oligomerization?

5. In Fig. 7, synergy between ABT263 and BDM19 or BTSA1 is shown in cells. ABT263 does not only disrupt the interaction between “BCL-XL” and “BAX”, but “BCL-XL” and “BIM” as well. Where is BIM in Fig. 7i? It is confusing that BIM is shown in Fig. S2c. Why the discrepancy between these two figures?

6. Would doxorubicin resistance of HCT116 P168G cells be removed by BDM19 (related to Fig. 2d)?

Minor points:

1. Please add carcinoma names beside the cell line notations in Fig. 1b.

2. Add notations to clarify the dominant BAX species (e.g. O and OO) in the various cell lines shown in Figures.

3. Fig. 1g; there is no positive control for anti-BAK antibody.

4. Fig. 2f; the SEC running buffer for this experiment is described as “IBc”, which only contains Tris/EGTA/sucrose (line 589). Is this correct? I would expect to see the presence of 0.5% CHAPS in the buffer especially for the mitochondria fraction, and SEC does not run well without salt (KCl).

Reviewer #4 (Remarks to the Author):

The authors performed virtual screening by screening 14 million small molecules referred to as an in-house dataset to a pharmacophoric model composed of putative interactions with the binding site (trigger site) on BAX protein. The active compounds were subsequently docked using induced-fit docking to the trigger site and to provide further insight into the impact of the active compound BDM19 on the conformation of BAX, the authors conducted molecular dynamics simulations for 1 microsecond.

To ensure the reproducibility of the computational workflow, some additional information needs to be provided in the Methods section (pages 31-32) as follows:

The authors should detail the parameters used in the software and provide the source code for the run scripts, except for commercial software used.

The dataset of 14 million compounds used for virtual screening should be described more comprehensively and made freely available to enable other researchers to validate the results of this study.

There is also a need to clarify why the dataset of purchasable compounds used for virtual screening, which the authors claim was obtained from eMolecules.com, is referred to as an in-house library on page 10, line 236. This discrepancy should be addressed to avoid confusion.

On page 16, line 382 (and in Supplementary Figure 6), the authors should quantify the increase in distance and give the average increase \pm standard deviation. This would give a clearer picture than the molecular dynamics trajectory presented as a distance versus time diagram.

Reviewer #5 (Remarks to the Author):

In this elegant study, Gitego et al. make significant progress towards understanding mechanisms of BAX activation and developing BAX activators by interrogating active and inactive monomers/dimers of BAX, following from their previous evidence for autoinhibited BAX dimers. They identify cancer cell lines in which resistance to BCL-2 inhibitors may be attributed to the stabilization of soluble inactive BAX dimers, consistent with BAX mutations found in refractile patient. Using molecular modeling to screen 14-million small molecule compounds that disrupt inactive BAX dimers, they identified BDM19 that binds the N-

terminal trigger site, resulting in an allosteric conformational change leading to BAX activation. This compound differs from other compounds identified thus far by this group and others despite the involvement of the same residues of BAX. Interestingly, treatment with BDM19 sensitized BAX dimer-enriched tumor cell lines compared to monomer-enriched cancer cells. Co-treatment of BDM19 with BCL-2/BCL-xL inhibitor ABT263 selectively sensitizes BAX dimer-enriched cell lines to undergo cell death. Thus, this study provides compelling evidence that inactive BAX dimers may serve as a reservoir of activatable BAX to induce apoptosis following BCL-2/BCL-xL inhibition.

Comments:

1. The effects of key BAX mutation K21E (R145E and R134E) on cell death induced by BDM19 appears to be missing. If lines 317 to 319 refer to such published cell viability data this was not clear, and requires readers to look up 4 cited papers (Refs 18,38,39,41) to decipher that K21E is impaired for staurosporine, BAM7, BTSA1, Eltrombopag, and BIM-SAHB-induced apoptosis, cyt c release and BAX oligomerization, but not the effect of K21E on BDM19.

2. Understanding the similarities and distinctions between the different BAX effectors is important to the field, and is mentioned throughout but is only partially summarized in the Discussion. Incorporation of these into a model or more complete discussion would be useful here (and/or subsequent commentary).

3. BAX monomer-dimer status of Namalwa and SUDHL-16 cells are not characterized in Fig. 1, which is important for drawing conclusions in Fig. 7 regarding susceptibility to BDM19. Not clear why the switch to these two cell lines for viability assays in 7a. Did the authors find exceptions to the rule for other cell lines in Fig. 1b?

4. Figure 4b needs clearer presentation of the findings, including addition of several residue numbers from 4a on blue CSPs in 4b, and marking at least a few other residues mentioned in the text (e.g. A24, L25, Q28, A46, D48, L45, L47, R109, Y164, T172, F176, W139). Also needed are the degrees of rotation left to right, and better indicated alpha 1 and 6, or other strategy to illustrate the trigger site pocket. A view from the BH3-binding side may also be useful to appreciate the trigger site shifts.

5. Several arguments support the statement that the shifted bands interpreted to be BAX dimers in Fig. 1 cannot be heterodimers with BCL-2 or BCL-xL (or BAK), at least in HCT116 and CALU-6 cells, as BCL-2, BCL-xL, and BAK are not present in the cytosolic fractions (they localize to mitochondria only or are not expressed to detectable levels). [Is this also true for the other cell lines used later in the manuscript (e.g. HPB-ALL, Namalwa, SUDHL-5/-16, OCI-AML3)?] Although the methods/buffers applied in Fig. 1b and 1c seem to be the same (Methods sections), the subsequent Western blot Methods where BCL-2 family antibodies are listed, describes a different lysis method (whole cells). It still may be informative to blot size exclusion fractions for multiple BCL-2 family and BH3-only proteins or other alternative methods to exclude heterodimers, or further clarify the text.

6. Although partially covered in the Discussion, it would be useful if the authors can comment further. Does the total level of cytosolic BAX differ between cell lines, or is there an informative correlation with the monomer:dimer ratio and any additional behaviors of monomers and inactive dimers, or localization of other BCL-2/BH3-only proteins, when comparing different cell lines that impinge on the propensity to form inactive dimers versus BAX monomers? Can BDM19 induce monomer formation in BAX P168G?

7. How do levels of BAK impact the effects of BDM19; all the blots shown have no BAK, perhaps by design? This could at least be mentioned. Given the wide usage of BAX/BAK DKO HCT116 cells, which seems pointless if there is truly no detectable BAK expression in HCT116 cells, or are BAX KO cells used here also DKOs?

8. The model in Figure S2c is not as transparent as Figure 7j. At first glance, figure is not intuitive; why draw “minor apoptosis” since the argument is that the cells are resistant to death? Step-wise BAX on left vs. right sides of Fig. S2c was not explained in the text or legend. Dissociation of inactive dimer is not directly depicted in either model; is their doubt that dissociation is needed prior to death?

9. In Figure 7a and 7e legend, indicate how “cell viability” was measured (celltiter glo, caspase3/7 activity).

10. Figure 4 legend is incorrectly labeled as Figure 3.

11. Figure key and significance of data presented in Fig. S5a-d or legend are not articulated.

12. Grammar error line 319.

We greatly appreciate all the reviewers' insights and valuable comments. To facilitate the review process, we indicated in each answer the figure panels or pages where the new results and text are displayed in the revised manuscript. Changes in the manuscript that address the reviewers' comments are marked in blue fonts.

Reviewer #1 (Remarks to the Author):

This manuscript described a small-molecule modulator, BDM19 that binds BAX, activates cytosolic dimers and prompts cells to BAX-mediated apoptosis alone or in combination with BCL-2/BCL-XL inhibitor Navitoclax. By using BDM19, the authors made many efforts to try to underscore the role of the cytosolic inactive BAX dimer in resistance to apoptosis of cancer cells and discussed the structure basis of Bax dimer activation. However, there are some concerns needed to be addressed. I recommend a major revision.

1. Since BTSA1 only activates Bax monomer rather than dimer, a solid structural basis of Bax dimer activation could have been achieved by comparing with BMD19. Do not only suggest that R134 and R145 are the hot spots because they are the additional residues bound by BMD19, but test the mutants. The identified new hot spot is the most valuable contribution in this study to the field of Bcl-2 family. Alternatively, based on the binding mode predicted by the authors, 3-Toluene and quinazoline of BDM19 could form cationic - π interaction with amino acid residues R145 and R134. Therefore, replacing aromatic group with molecules that cannot form a connection - π interaction with R145 and R234, such as 3-methylepoxyde or tetrahydroquinazoline, could verify whether R145 and R134 are the key amino acid residues for Bax dimer activation.

We thank the reviewer for the positive remarks and experimental suggestions to strengthen our conclusions. Indeed, R134 and R145 residues are two residues that are predicted to interact with BDM19 and not with BTSA1 and experimentally we showed by MST binding assay that R134E and R145E mutants diminish the interaction with BDM19 (**Fig. 5a**). We assumed that the reviewer would like us to test these mutations in cells evaluating the interaction of BDM19 with these BAX mutants. For this, we transduced MEFs BAX/BAK DKO cells to stably express human BAX R134E and BAX R145E mutants, tested them and compared them with MEF DKO BAX WT and BAX K21E cells. We compared these cell lines upon BDM19 treatment using the BN-PAGE and 6A7 assays for activation of the BAX dimer, the cellular thermal shift assay for cellular BAX engagement and the caspase 3/7 activation assay for apoptosis induction. Interestingly, BN-PAGE assay (**Suppl. Fig. 6a**) showed that R134E and R145E mutations disrupt the cytosolic BAX dimer and have a cytosolic BAX monomer conformation. This can be anticipated since R134 and R145 participate in intermolecular interactions within the inactive BAX dimer based on the crystal structure previously reported (Garner et al. Mol. Cell., 2016). BN-PAGE showed that BDM19 reduced cytosolic BAX WT dimer to active monomer however, BDM19 does not induce activation of cytosolic BAX R134E or BAX R145E monomers (**Supp. Fig. 6a**). Cellular thermal shift assays showed that BDM19 had a significant effect in decreasing the melting point of BAX WT, however, it had a diminished effect with BAX R134E, R145 and K21E mutants, suggesting only effective target engagement of BDM19 with BAX WT (**See figure for reviewers at the end of the document, page 15**). BDM19 treatment alone had no apoptotic effect in MEFs presumably from high expression of anti-apoptotics, but with a sublethal dose of Navitoclax, BDM19 induced apoptosis and BAX activation in MEF BAX WT cells, as probed by the caspase-3/7 and 6A7 assays (**Suppl. Fig. 7a-b**). However, in MEFs expressing the BAX R134E, R145 and K21E

mutants, BDM19 showed no or significantly weaker activity in apoptosis induction and BAX activation, as probed by the caspase-3/7 and 6A7 assays (**Suppl. Fig. 7a-b**). Taken together, our studies with various binding and activity assays (MST, BN-PAGE, CETSA, 6A7 IP and Caspase-3/7) provide consistent evidence that residues R134E, R145 and K21E, are critical for the interaction and activity of BDM19 to BAX. Moreover, BDM19 interaction with these residues is supported by our NMR and molecular docking studies.

To further support the binding mode of BDM19 with BAX, we also synthesized analogs of BDM19 to modify the predicted interactions with residues R145 and R134. Unfortunately we couldn't synthesize the methylepoxy analog suggested by the reviewer because of the difficulty of the synthesis of the molecule. Instead, we replaced the 3-toluene of BDM19 with cyclopentyl in BDM19.2 and cyclopropyl in BDM19.3 and the quinazoline ring in BDM19 with the tetrahydroquinazoline in BDM19.4 (see **Suppl. Fig. 5c** for structures of the molecules). These modifications should eliminate the predicted aromatic interactions with R134 and R145 that were proposed from the docking studies. Indeed, binding assays with MST showed diminished binding to BAX from BDM19.2, BDM19.3 and BDM19.4 analogs, in contrast to BDM19 binding (**Fig. 5d**). Moreover, treatment of Calu-6 cells with the compounds and 6A7 co-immunoprecipitation, revealed that only BDM19, and none of the analogs BDM19.1, BDM19.2, BDM19.3 and BDM19.4, is capable of inducing 6A7 epitope exposure and activation of BAX (**Fig. 5e**). Therefore, data with the BDM19 analogs support the specificity of BDM19 to BAX in vitro and in cells and support the predicted binding mode and interactions with R134 and R145 (**Fig. 5b,c**) from the NMR/molecular docking studies.

2. The Bax activation by BDM19 in Figure 7g is not concentration-dependent. It challenges BDM19's Bax activation.

We repeated again this experiment to evaluate BAX activation at 2 hrs aiming to match the concentrations of the drugs in the viabilities assay where there is synergistic activity (Fig. 7e,f). The result (**Fig. 7g**) shows BDM19-induced BAX activation of BAX dimers in Calu-6 cells in combination with ABT-263 that is concentration dependent and more potent compared to BTSA1 in the corresponding experiment, as suggested by the viability assays (Fig. 7e,f).

3. NMR, MD and FP demonstrated that the binding sites of BDM19, BTSA1 and Bim are highly overlapped. All these compounds bind in the groove between $\alpha 1$ and $\alpha 6$, and they all interact with residues K21, M20, M137 and L141. As such, it is not proper to claim that BMD19 has a distinct binding site with other Bax activators.

We agree with the reviewer that BDM19 and BIM peptide and other BAX activators have overlapping interactions with BDM19. For the binding of BDM19 we used statements in the text such as "unique/distinct binding interaction" or "unique binding mode" to characterize the binding of BDM19, not the binding site as all these molecules bind to the BAX trigger site. In any case, we modified the text to be more precise and refer to a distinct binding mode/interaction of BMD19 compared to other BAX small molecules. We have also introduced **Figure 8b** to show the distinct

binding mode at the BAX trigger site in comparison to BTSA1 and eltrombopag molecules and discussion in the text.

4. Since MD result suggested that BDM19 induces Bax conformation change and activation similar as BTSA1, why BMD19 cannot induce Bax monomer and apoptosis in OMI-AML3 cell lines? Why the Bax monomer activation by BMD19 is weaker than BTSA1?

Indeed, this is an interesting finding of our work and we will try to explain the difference between BTSA1 and BDM19 and how these may suggest the preference for BAX monomer by BTSA1 activator and for BAX dimers by BMD19. BTSA1 has more potency to compete BIM BH3 binding to the BAX trigger site in FPA assay, IC₅₀ = 200 nM (Reyna et al, Cancer Cell 2017) compared to BDM19, IC₅₀ = 5 μM (Fig. 3f). By MST binding approach, we can get a binding curve to BAX of BDM19, K_d = 500 nM (Fig. 5a) but with BTSA1 we can't get a reasonable binding curve because of the potency of the molecule to conformationally activate BAX even at low nM concentrations. Thus, BTSA1 is a better activator than BDM19 for BAX monomer used in these binding assays. BDM19 is able to separate and activate the BAX dimer in BN-PAGE (**Fig. 3c-d**) and by 6A7 assay in Calu-6 cells (**Fig. 5e and Fig. 7g**), whereas BTSA1 cannot separate the BAX dimer to BAX monomers (**Suppl. Fig. 4b**) or activate the BAX dimers (**Fig. 7g**) with the same efficiency as BDM19. The strategy for *in silico* design/screening for each compound was different, as for the BTSA1 it focused on the BAX monomer structure at the trigger site around residue K21 and for BDM19 it focused on the BAX dimer structure interface that included also additional residues of helix α6 such as R134 and R145. As we describe in the text, our docking studies also suggest that the two molecules have a different binding orientation in the BAX trigger site consistent with the *in silico* design.

To further compare the two compounds, we performed similar MD simulations of BTSA1 bound to BAX monomer, and we found that BTSA1 in comparison to BDM19 affected the same residues and interfaces but in several measured distances BTSA1 induced larger changes, which may suggest a better potency to activate the BAX monomer (**Fig 6, Suppl. Fig. 8-10**). Particularly, several residues of the α1- α2 loop moving further away from helix α6 in contrast to the inactive structure and the BDM19-BAX structure, maybe the reason that BTSA1 induces better activation of the BAX monomer (**Suppl. Fig. 8e-h**). We have previously found that the α1- α2 loop opening is a key conformational change for BAX activation which also coincides with the exposure of the 6A7 epitope (Gavathiotis et al Mol. Cell, 2010, Garner et al, Mol. Cell. 2016, Spitz et al. Nat. Cmmun 2021). BTSA1 is modelled to interact closely with residues at helix α1 near the α1- α2 loop within the BAX trigger site and it may induce these changes better (Reyna et al, Cancer Cell 2017).

Therefore, we suggested in the discussion text that the specific binding orientation of BDM19 and interactions with residues of helix 6 within the BAX trigger site as assessed in this revised article, may reduce the activation efficacy of BDM19 for the BAX monomer compared to BTSA1. Importantly, these residues R134 and R145 on helix 6 that BDM19 interacts with, are key to the binding and inhibition of BAX activation by eltrombopag. To further clarify these points we also provide a structural model (**Fig 8b**) and figure legend to describe differences among the BTSA1 and BDM19 interactions with BAX trigger site.

5. The author run 1000 ns in MD but RMSD still indicates that the system did not reach the equilibrium phase. The reliability of MD is weakened.

We respectfully disagree with the assessment that our molecular dynamics simulations are weakened. We provide new data for radius of gyration and total energy plots in addition to RMSD plots (**Supp. Fig 8a-d**) which show that the systems in the MD simulations have reached equilibrium phase. The differences in RMSD, particularly in the presence of BDM19, suggest local conformational changes of the BAX structure consistent with its conformational activation mechanism and this is what we analyzed and reported from the MD simulations by the various plots and distance measurements (**Fig 6, Suppl. Fig. 8-10**). We have also added 3 additional simulations under the same conditions of the BTSA1-BAX complex (**Suppl. Fig. 8-10**).

6. Two cell lines (SUDHL-16 and Namalwa) used for cell experiments in Figure 7 are not in Figure 1, so the level of Bax dimer and monomer there is unclear. This reviewer suggests the author use the cell lines in Figure 1.

We agree with the reviewer. We performed SEC analysis for the two cell lines SUDHL-16 and Namalwa (**Fig. 1b**) and found these cell lines express BAX dimers and monomers.

7. It is strange Eltrombopag could improve the dynamic of Bax conformation like BMD19 as the author claimed at Page7, Line402. It is established that the Bax inhibitor Eltrombopag stabilizes the inactive conformation of Bax.

We agree with the reviewer. We corrected and revised the text.

8. Please provide the NMR, HPLC and MS data and spectrum of the final compound BMD19.

We now provide analytical data of BDM19 and its analogs along with their chemical syntheses in supplemental methods and the corresponding ¹H NMR spectra in **Supp. Fig. 12-16**

9. Many editing mistakes need to be improved.

We reviewed thoroughly the text and made several improvements and corrections. Thank you.

Reviewer #2 (Remarks to the Author):

The authors present a wonderful story about the role of cytosolic inactive Bax monomers and dimers in avoiding cancer by tumor cells and an elegant way of activation those species by small effector ligands to become active and overcome apoptotic hindrance. In this nice piece of work they also highlight many aspects of the regulation of those species and of activated Bax in various cells lines and how those species are linked to each other and their apoptotic activity upon translocation to the mitochondria for membrane perforation and blocking anti-apoptotic proteins like Bcl-2 e.g. to protect cells against cell death. In their well designed and executed study they use a wide range of powerful methods starting from cell assays to in silico drug screening, NMR and other binding assays and even MD simulations. And indeed, the results and how they are presented is very impressive.

From my point of perspective there are only a few minor points the authors show consider.

We thank the reviewer for the positive remarks of our work and appreciate his/her suggestions to improve our manuscript.

1. They authors used a range of cancer cell lines. I wonder how they have selected them and if it would have been possible to use a non-cancer cell line reflecting normal cells for comparison e.g. Many cells who undergo apoptosis are not cancer cells but normal cells e. g. like under embryogenesis.

Our choice of cell lines is based on a selection of various cancers with diverse oncogenes and mutations among the cell lines to avoid any potential bias. We now provide the analysis for BAX monomers and dimers by SEC of two additional human, non-cancerous cell lines, human epithelial BEAS-2B cells and human lung fibroblasts IMR90 cells (**Figure 1b**). We previously analyzed mouse embryonic fibroblasts (MEFs) by SEC (Garner et al. Mol. Cell 2016). Interestingly, SEC analysis suggests that both BEAS-2 and IMR90 cells express BAX dimers.

2. They authors used Navitoclax which inhibits Bcl-2/Bcl-XL. I wonder if the authors tried also the Bcl-2 specific Venetoclax inhibitor or would Bcl-xl have compensated for that in the authors assays.

We had provided caspase-3/7 activation assay data (**Suppl. Fig. 3a-b** in the revised manuscript) of HCT116 cell lines upon Venetoclax and BCL-XL specific inhibitor A-1331852 treatment and found that the results with Navitoclax are also reproduced with the BCL-XL specific inhibitor A-1331852. However, Venetoclax treatment lacks apoptotic activity due to limited expression of BCL-2 protein and cells primarily relying on BCL-XL for inhibition of the mitochondrial apoptotic pathway (**Fig. 1g**).

3. What was the concentration of the Bax samples used in NMR measurements and what was the yield of the ¹⁵N-labelled Bax by their overexpression systems used?

We had the concentration of NMR samples in the NMR methodology and we also added the yield of ¹⁵N-labeled BAX in the same paragraph, page 38.

4. In the introduction the authors mentioned Bax mutations P168A and G67R found in AML patients and interfering with Venetoclax. Since there are also mutation G101V (1) found in Bcl-2 preventing Venetoclax working properly, I wonder if the authors might have a molecular idea about how Bax mutants interfere there. I always thought (quite simplified probably) that Venetoclax binds to Bcl-2 and Bax gets just released to do its work. But this might be too simple thought by me.

We are happy to comment further on this. Indeed, the reviewer is correct that Venetoclax works by binding to BCL-2 to release BAX, therefore mutations of BCL-2 like G101V impair Venetoclax binding to BCL-2 and its ability to release BAX or bound BH3-only proteins. BAX mutants such as P168A and G67R (Moujalled et al, Blood, 2022, Meijerik et al, Blood, 1998) have impaired translocation to mitochondria and therefore they do not form complex with the BCL-2 protein on the mitochondria outer membrane. Our work in this manuscript also shows that HCT116 BAX

P168G remains a dimer and cytosolic protein. In this case, Venetoclax binding to BCL-2 does not lead to release of activated BAX and induction of apoptosis. Moreover, if cells are not primed and BH3-only proteins such as BIM are not bound to BCL-2 protein at sufficient amounts, then Venetoclax is challenged to activate BAX mutants such as P168A and G67R that can form cytosolic dimers and seem to be more resistant than BAX WT to BIM-induced BAX activation. Therefore, these BAX mutations can cause resistance to Venetoclax.

5. Line 223 “cytosolic inactive Bax dimer constitutes a road block to Bax activation..”

Do the authors know why apoptotic regulation would need this kind of mechanism: how are the levels of inactive Bax dimers in the cell lines used by the authors. And how are those levels compared to non-cancer cell lines (what feeling do the authors have about those levels; does not need to have been measured before)?

We thank the reviewer, this is an interesting question. We attempted to address this by quantifying BAX protein levels and measuring apoptotic priming of cell lines using the BH3 profiling approach. We now provide additional data and plot the correlations of total protein levels of BAX with BAX dimers in cell lines used in this study (**Supp. Fig. 1c**), and the correlation between protein levels of BAX dimers and monomers with the degree of apoptotic priming as determined by the percent of depolarization by the BIM BH3 peptide (**Suppl. Fig. 2a**). We found positive trends between the amount of total protein levels of BAX and BAX dimers and increased amount of BAX dimers with low apoptotic priming of the cells. These data suggest that cells have propensity for BAX dimers when higher levels of BAX are expressed and when cells are less primed to mitochondrial apoptosis. Less primed mitochondria is linked to low levels of BH3-only proteins bound to anti-apoptotic BCL-2 proteins at the mitochondrial membrane. The precise mechanism for the formation of BAX dimers is not clear to us but we suggest that it can be a regulatory mechanism to control sensitivity to apoptosis and priming of cells. Higher levels of activated BH3-only proteins such as BIM, increase priming of cells to apoptosis, and in this process BAX dimers can shift to BAX monomers as BIM BH3 can disrupt BAX dimers to monomers. Therefore, cells with higher protein levels of activated BH3-only proteins neutralized by anti-apoptotic BCL-2 proteins and more BAX monomers would be in a higher primed state and more sensitive to pro-apoptotic treatments. In contrast, BAX cytosolic dimers would be present in cell with a lower primed state and associated with resistance to apoptosis and survival. Although we have analyzed a small number of non-cancerous cell lines, we found that these cell lines express BAX dimers and are less primed for apoptosis, which is consistent with what we found in cancer cell lines. We have provided additional text to describe these results and in the discussion.

6. Lines 324-339: very detailed molecular description: could be shortened I believe.

We think it is necessary to explain in detail the molecular interactions and binding experiments, particularly since other reviewers suggested the importance of these interactions in the distinct activity of BDM19 from other BAX activators. However, we attempted to shorten the text.

7. Line 505...: What happens to the BDM19 ligand after binding to inactive Bax dimers and converting them into active ones who can trigger apoptosis. Is the ligand losing its affinity upon

conversion of Bax and if not is Bax then carry out mitochondrial outer membrane formation with the ligand attached to it (could Bcl-2 bind activated Bax with BDM19 ligand attached):

BAX activation by BH3 peptides and small molecules has been shown to be a hit and run mechanism (reviewed in Spitz et al. Trends Pharmacol. Sci. 2022). Binding of the peptide/small molecule triggers conformational activation of BAX enabling BAX to translocate, insert into the outer mitochondria membrane and oligomerize to induce MOMP. Continuous binding of these peptide/small molecule is not possible when BAX has transformed its conformation, most likely the peptide/small molecule would interfere with the process of conformational activation and oligomerization. To evaluate whether BDM19 stays bound to BAX when it adopts an active conformation that can proceed to mitochondrial outer membrane integration/oligomerization, we performed binding of BDM19 with detergent activated BAX, which has been established to generate active and functional BAX in MOMP assays. BDM19 was found not capable to bind to detergent activated BAX (**Supp. Fig. 5d**) suggesting that BDM19 loses its affinity upon BAX conformational activation. We previously showed that BDM19 binds inactive BAX with a K_d of 560 nM using the same MST binding assay.

Reviewer #3 (Remarks to the Author):

Gitego et al. report their new research on a pro-apoptotic protein, BAX, extended from the prior work from the Gavathiotis laboratory concerning the inactive form of BAX dimers. They found that BAX can form a dimer that makes it resistant to its activation. Now, they report that this form of BAX is found in the cytosolic fraction of many cancer cell lines and that these cells are more resistant to apoptosis inducers. Anti-cancer therapies targeting BAX are a promising approach, however, the presence of these inactive BAX dimers in the cells would hinder it. In order to probe the mechanism and develop a means to reduce these inhibitory dimers, they screened a compound library and discovered BDM19, which disrupts the inactive BAX dimers. Using structural biological techniques, they show that BDM19 interacts with the BAX trigger site in a different manner from that of BIM BH3 or a BAX activator compound, BTSA1 they previously reported.

The study is carefully conducted and in general the data are of high quality. It would give yet new insights to BAX activation processes as well as new tools to the field of cancer research. In order to make the results more convincing and useful to the field, I would like the following issues to be addressed/clarified:

We thank the reviewer for the positive remarks of our work and appreciate his/her suggestions to improve our manuscript.

1. The authors should describe Western blotting of BN-PAGE in more detail. BN-PAGE takes advantage of Coomassie Blue binding to the proteins to cover the charge residues. Such proteins lose the charge, therefore they are not transferred to the PVDF/nitrocellulose electronically. It is unclear how this hurdle was overcome in the method. Moreover, it is surprising that transferred proteins did not seem to be denatured and yet anti-BAX antibody detected it. These issues should be clarified, as this method would be useful when other investigators wish to detect the inactive BAX dimers who are not necessarily equipped with the

more elaborate SEC system.

We thank the reviewer for the opportunity to clarify the BN-PAGE assay in more detail and we provide further clarifications in the revised methods. The Coomassie blue G-250 is an anionic dye, which binds proteins and confer them a negative charge allowing them to migrate to the anode during electrophoresis. Additionally, the cathode buffer additive in the dark and light cathode buffers contains coomassie blue G-250 to maintain the negative charge of proteins throughout the BN-PAGE run. After electrophoresis, the gels are equilibrated in the transfer buffer (1X NuPage transfer buffer with 20% methanol) supplemented with 0.037% SDS for 10 minutes to ensure protein denaturation, a negative charge on proteins and an effective transfer on the PVDF membrane. The anti-BAX antibody (Cell signaling 2772) binds to the unstructured N-terminal region (aa 1-14) of BAX which is exposed in the soluble form of BAX either in the monomer (Suzuki et al. Cell 2000) or dimer conformation (Garner et al. Mol. Cell 2016). So BAX in monomer or dimer conformation is detected in BN-PAGE without the requirement for denaturation.

2. Related to the above, would there be a more convenient way to detect these dimers, using the whole cell lysate without cell fractionation? BAX would be oligomerized if non-ionic detergent such as TX-100 was used to lyse cells, so how about CHAPS or digitonin? Would the use of these detergents be compatible with the detection of the inactive dimers?

We used cell fractionation because we wanted to focus on investigating cytosolic BAX. As a more convenient way to analyze the conformation of cytosolic BAX we introduced the BN-PAGE assay instead of SEC, which only requires treatment of the cells with a digitonin buffer and a centrifugation step to isolate the cytosolic fraction for subsequent BN-PAGE analysis.

3. How would the authors explain the “in-between” size shown in OCI-AML3 cell lysate (Fig. 1b) and HCT116 E75K (1f)? Also, in Fig. 2f, the peak size of BAX in WT cytosol and mitochondria and P168G cytosol appears “in-between” the monomer and dimer, not monomeric, as the authors interpreted (lines 214-219). Please explain.

We thank the review for the opportunity to clarify this. Indeed in some cases the SEC analysis of cytosolic BAX suggests that BAX protein has a “in-between” size between a monomer and a dimer. This has been also observed in purifications of recombinant BAX by SEC when the gel filtration column is saturated because of the high protein amount. Therefore, this effect can also vary by the concentration/expression levels of BAX in each cytosolic fraction and the capacity of gel filtration column to separate many other proteins of similar size in the cell extract. We cannot exclude other cellular factors although our proteomics studies have not identified any specific BAX modification. Indeed, for **Fig. 2f**, we mentioned in the text that, after treatment with ABT-263, some BAX WT and P168G dimers are reduced to monomers because of the shift of the BAX bands to lower molecular weight from the BAX dimers observed in the untreated cells (Fig. 1f).

4. In Fig. 2, various BAX mutants expressing cells were compared in their sensitivity to apoptosis. Do the authors know that BAX P168G, if rendered monomeric, has the same activity as WT? In other words, P168G mutation does not affect the series of changes BAX undergoes during activation, such as conformational change, insertion of $\alpha 9$, forming BH3 swapped dimers and

oligomerization?

If BAX P168G mutant is rendered monomeric, as this is shown to some extent with the treatment of cells with ABT-263 (**Fig. 2f**), it is still challenging for BAX to translocate to the mitochondria and proceed to MOMP as BAX P168G mutation inhibits the movement of the c-terminal helix and effects the ability of the BAX protein to translocate and insert at the mitochondrial outer membrane (Gavathiotis et al, Mol. Cell 2010, Garner et al. Mol. Cell 2016). Therefore, ABT-263 is more effective and induced BAX activation, translocation and oligomerization in HCT116 WT cells (**Fig. 2f**). Interestingly, our new data suggest that BDM19 in combination with doxorubicin is capable to induce apoptosis in HCT116 BAX P168G cells, albeit to a lesser degree than BAX WT (**Suppl Fig. 11b**). This result suggests that BDM19 with the increased priming induced by doxorubicin is capable to induce apoptosis by BAX P168G.

5. In Fig. 7, synergy between ABT263 and BDM19 or BTSA1 is shown in cells. ABT263 does not only disrupt the interaction between “BCL-XL” and “BAX”, but “BCL-XL” and “BIM” as well. Where is BIM in Fig. 7i? It is confusing that BIM is shown in Fig. S2c. Why the discrepancy between these two figures?

We agree with the reviewer, BIM was unintentionally omitted from Figure 7i. We added BIM to the new model in Figure 8a and clarify the model further in the figure legend.

6. Would doxorubicin resistance of HCT116 cells be removed by BDM19 (related to Fig. 2d)?

We thank the reviewer for this suggestion. We used the combination of doxorubicin and BDM19 in HCT116 BAX P168G cells (**Suppl Fig. 11b**) and we found that the combination of the two compounds can overcome the resistance due to the BAX P168G mutation. Although induction of apoptosis reaches higher degree in HCT116 BAX WT cells compared to HCT116 BAX P168G cells, we think the result is promising for the combined activity of doxorubicin and BDM19 compound to address a challenging mutation to activate such as P168G/A which has been shown to provide resistance to Venetoclax treatment in patients (Moujalled et al, Blood, 2022)

Minorpoints:

1. Please add carcinoma names beside the cell line notations in Fig. 1b.

We added carcinoma names in Fig. 1b.

2. Add notations to clarify the dominant BAX species (e.g. O and OO) in the various cell lines shown in Figures.

We have added notations in figures where appropriately to clarify the dominant BAX species.

3. Fig. 1g; there is no positive control for anti-BAK antibody.

We have used a new anti-BAK antibody and a positive control and we detected the expression of BAK in HCT116 cell lines (**Fig. 1g, Suppl. Fig. 2b**).

4. Fig. 2f; the SEC running buffer for this experiment is described as “IBc”, which only contains Tris/EGTA/sucrose (line 589). Is this correct? I would expect to see the presence of 0.5% CHAPS in the buffer especially for the mitochondria fraction, and SEC does not run well without salt (KCl).

The description of IBc buffer is correct. 0.5% CHAPS is used to lyse the mitochondria fraction but it is not necessary to be included in the SEC running buffer. We used Tris HCl that acts as salt that buffers the solution and we did not see any difficulty running the samples with SEC. We provide these details in the methodology description.

Reviewer #4 (Remarks to the Author):

The authors performed virtual screening by screening 14 million small molecules referred to as an in-house dataset to a pharmacophoric model composed of putative interactions with the binding site (trigger site) on BAX protein. The active compounds were subsequently docked using induced-fit docking to the trigger site and to provide further insight into the impact of the active compound BDM19 on the conformation of BAX, the authors conducted molecular dynamics simulations for 1 microsecond.

To ensure the reproducibility of the computational workflow, some additional information needs to be provided in the Methods section (pages 31-32) as follows:

We thank reviewer’s suggestions for additional information to ensure reproducibility.

The authors should detail the parameters used in the software and provide the source code for the run scripts, except for commercial software used.

Only commercial software (Schrodinger, LLC) were used for various computational approaches. No source code or scripts are used, default parameters of Schrodinger software were selected for each Schrodinger module, and any exceptions in parameter selection is provided in the methods description. We added more details about the pharmacophore screen and MD simulations parameters.

The dataset of 14 million compounds used for virtual screening should be described more comprehensively and made freely available to enable other researchers to validate the results of this study.

We appreciate reviewer’s request for the reproducibility but the library of molecules is provided by a commercial source eMolecules Inc. and we have no rights to distribute it or upload it in public domain. However, eMolecules Inc. (www.emolecules.com) as it is mentioned on their website, can provide the library to any researcher requesting it.

There is also a need to clarify why the dataset of purchasable compounds used for virtual screening, which the authors claim was obtained from eMolecules.com, is referred to as an in-house library on page 10, line 236. This discrepancy should be addressed to avoid confusion.

We agree, our description of “in-house” is confusing as we refer to the same eMolecules library after processing it with the LIGPREP and EPIK software (Schrodinger, LLC) as described in the Methods. We deleted the description ‘in-house’ in the text.

On page 16, line 382 (and in Supplementary Figure 6), the authors should quantify the increase in distance and give the average increase +/- standard deviation. This would give a clearer picture than the molecular dynamics trajectory presented as a distance versus time diagram.

We quantified and provided average increase +/- standard deviation for all distances related to molecular dynamics simulations in **Supp. Figure 8-10**. We also did that for the new MD simulations with BTSA1-BAX complex.

Reviewer #5 (Remarks to the Author):

In this elegant study, Gitego et al. make significant progress towards understanding mechanisms of BAX activation and developing BAX activators by interrogating active and inactive monomers/dimers of BAX, following from their previous evidence for autoinhibited BAX dimers. They identify cancer cell lines in which resistance to BCL-2 inhibitors may be attributed to the stabilization of soluble inactive BAX dimers, consistent with BAX mutations found in refractile patient. Using molecular modeling to screen 14-million small molecule compounds that disrupt inactive BAX dimers, they identified BDM19 that binds the N-terminal trigger site, resulting in an allosteric conformational change leading to BAX activation. This compound differs from other compounds identified thus far by this group and others despite the involvement of the same residues of BAX. Interestingly, treatment with BDM19 sensitized BAX dimer-enriched tumor cell lines compared to monomer-enriched cancer cells. Co-treatment of BDM19 with BCL-2/BCL-xL inhibitor ABT263 selectively sensitizes BAX dimer-enriched cell lines to undergo cell death. Thus, this study provides compelling evidence that inactive BAX dimers may serve as a reservoir of activatable BAX to induce apoptosis following BCL-2/BCL-xL inhibition.

We thank the reviewer for the positive remarks of our work and appreciate his/her comments and suggestions to improve our manuscript.

Comments:

1. The effects of key BAX mutation K21E (R145E and R134E) on cell death induced by BDM19 appears to be missing. If lines 317 to 319 refer to such published cell viability data this was not clear, and requires readers to look up 4 cited papers (Refs 18,38,39,41) to decipher that K21E is impaired for staurosporine, BAM7, BTSA1, Eltrombopag, and BIM-SAHB-induced apoptosis, cyt c release and BAX oligomerization, but not the effect of K21E on BDM19.

We agree with reviewer that our statement is not so clear. However, in that section, we referred to in vitro data for binding and BAX activation of the previous BAX binders in the literature as we showed that BDM19 has impaired binding to K21E, R134E, R145 mutants by MST assay. We now also provide data to compare the effects of BDM19 with DKO MEFs expressing BAX WT, K21E, R134E, R145E for i) BAX activation with the 6A7 co-immunoprecipitation assay, ii) cellular BAX engagement with the cellular thermal shift assay and iii) BAX-induced apoptosis with the caspase-3/7 activation assay. We found that BDM19 treatment of DKO MEFs expressing various BAX variants had impaired cellular BAX engagement (**See figure for reviewers at the end of**

the document, page 15), BAX-induced apoptosis and BAX activation (**Suppl. Fig. 7a,b**) in the presence of BAX K21E as well as BAX R134E and BAX R145E compared to BAX WT. A similar comment #1 was also addressed for reviewer 1.

2. Understanding the similarities and distinctions between the different BAX effectors is important to the field, and is mentioned throughout but is only partially summarized in the Discussion. Incorporation of these into a model or more complete discussion would be useful here (and/or subsequent commentary).

We thank the reviewer for this suggestion. We have now incorporated the various BAX trigger site compounds into a model using the surface of the BAX trigger site (**Fig. 8b**). We also provided information in the legend about their distinct binding mode and interactions. We have also provided additional results (MD simulations of BTSA1-BAX complex (**Supp. Fig.8-10**) and discussion to inform about the similarities and differences of the BAX effectors.

3. BAX monomer-dimer status of Namalwa and SUDHL-16 cells are not characterized in Fig. 1, which is important for drawing conclusions in Fig. 7 regarding susceptibility to BDM19. Not clear why the switch to these two cell lines for viability assays in 7a. Did the authors find exceptions to the rule for other cell lines in Fig. 1b?

We unintentionally omitted similar analysis of Namalwa and SUDHL-16 cells in Figure 1b to characterize the BAX monomer-dimer status. We agree that is an important analysis to draw conclusions for our data in Fig. 7 and activity of BDM19. We performed SEC analysis for the two cell lines SUDHL-16 and Namalwa (**Fig. 1b**) and found these cells express BAX dimers and monomers. We have not find cell lines that suggest exception to the rule.

4. Figure 4b needs clearer presentation of the findings, including addition of several residue numbers from 4a on blue CSPs in 4b, and marking at least a few other residues mentioned in the text (e.g. A24, L25, Q28, A46, D48, L45, L47, R109, Y164, T172, F176, W139). Also needed are the degrees of rotation left to right, and better indicated alpha 1 and 6, or other strategy to illustrate the trigger site pocket. A view from the BH3-binding side may also be useful to appreciate the trigger site shifts.

We have improved the presentation of the NMR findings in **Fig. 4b** by labeling residues with significant CSPs in 4a that cover the trigger site (helices 1 and 6). We illustrated the trigger site on the BAX surface with CSPs to demonstrate their overlap. A direct view of the canonical BH3 groove is not possible as BAX has its C-terminal helix occupying this BH3 groove but in **Fig. 4c**, we show CSPs that extend to the C-terminal helix and residues of helix 5 and 4. We also provided labeling of the two different views in **Fig. 4b and 4c**.

5. Several arguments support the statement that the shifted bands interpreted to be BAX dimers in Fig. 1 cannot be heterodimers with BCL-2 or BCL-xL (or BAK), at least in HCT116 and CALU-6 cells, as BCL-2, BCL-xL, and BAK are not present in the cytosolic fractions (they localize to mitochondria only or are not expressed to detectable levels). [Is this also true for the other cell lines used later in the manuscript (e.g. HPB-ALL, Namalwa, SUDHL-5/-16, OCI-AML3)?] Although the methods/buffers applied in Fig. 1b and 1c seem to be the same (Methods sections), the subsequent Western blot Methods where BCL-2 family antibodies are listed, describes a different lysis method (whole cells). It still may be informative to blot size exclusion

fractions for multiple BCL-2 family and BH3-only proteins or other alternative methods to exclude heterodimers, or further clarify the text.

We performed western blot analysis and fractionation of cytosolic and mitochondrial fractions of two additional cell lines, Namalwa and SUDHL-16 and found that anti-apoptotic BCL-2, BCXL, MCL-1, BIM_{EL}, BIM_S and BAK are found in the mitochondrial fraction with the exception that some BCL-XL is detected also in the cytosolic fraction of SUDHL-16 (**Suppl. Fig. 1a**). We performed size exclusion chromatography and western blot analysis of the cytosolic fraction of SUDHL-16 cells and we detected elution of BCL-XL at a higher molecular weight from the BAX dimer and not at the same fractions as the BAX dimer (**Suppl. Fig. 1b**). These additional data in Namalwa and SUDHL-16 cell lines and in Calu-6 and HCT116 cell lines in this paper, as well as previously in MEFs (Garner et al. Mol. Cell 2016), excluded the possibility of the presence of cytosolic BAX heterodimer with anti-apoptotic BCL-2 and BIM proteins.

6. Although partially covered in the Discussion, it would be useful if the authors can comment further. Does the total level of cytosolic BAX differ between cell lines, or is there an informative correlation with the monomer:dimer ratio and any additional behaviors of monomers and inactive dimers, or localization of other BCL-2/BH3-only proteins, when comparing different cell lines that impinge on the propensity to form inactive dimers versus BAX monomers? Can BDM19 induce monomer formation in BAX P168G?

We thank the reviewer, these are interesting comments. We attempted to address this by quantifying BAX protein levels and measuring apoptotic priming of cell lines using the BH3 profiling approach. We now provide additional data and plot the correlations of total protein levels of BAX with BAX dimers in cell lines used in this study (**Supp. Fig. 1c**), and the correlation between protein levels of BAX dimers and monomers with the degree of apoptotic priming as determined by the percent of depolarization by the BIM BH3 peptide (**Suppl. Fig. 2a**). We found positive trends between the amount of total protein levels of BAX and BAX dimers and increased amount of BAX dimers with low apoptotic priming of the cells. These data suggest that cells have propensity for BAX dimers when higher levels of BAX are expressed and when cells are less primed to mitochondrial apoptosis. Less primed mitochondria is linked to low levels of BH3-only proteins bound to anti-apoptotic BCL-2 proteins at the mitochondrial membrane. The precise mechanism for the formation of BAX dimers is not clear to us but we suggest that it can be a regulatory mechanism to control sensitivity to apoptosis and priming of cells. Higher levels of activated BH3-only proteins such as BIM, increase priming of cells to apoptosis, and in this process BAX dimers can shift to BAX monomers as BIM BH3 can disrupt BAX dimers to monomers. Therefore, cells with higher protein levels of activated BH3-only proteins neutralized by anti-apoptotic BCL-2 proteins and more BAX monomers would be in higher primed state and more sensitive to pro-apoptotic treatments. In contrast, BAX dimers would be present in cells with a lower primed state and associated with resistance to apoptosis and survival. Although we have analyzed a small number of non-cancerous cell lines, we found that these cell lines express BAX dimers and are less primed for apoptosis, which is consistent with what we found in cancer cell lines. We have provided additional text to describe these results and in the discussion.

Regarding BDM19 and BAXP168G mutant, we have added new data that suggest that BDM19 in combination with doxorubicin is capable to induce apoptosis in HCT116 BAX P168G cells,

albeit to a lesser degree than BAX WT. This result suggests that BDM19 with the increased priming induced by doxorubicin is capable to activate BAX P168G and induce apoptosis.

7. How do levels of BAK impact the effects of BDM19; all the blots shown have no BAK, perhaps by design? This could at least be mentioned. Given the wide usage of BAX/BAK DKO HCT116 cells, which seems pointless if there is truly no detectable BAK expression in HCT116 cells, or are BAX KO cells used here also DKOs?

We repeated the western blot detection of BAK, as requested by another reviewer, with a new anti-BAK antibody using control cells and HCT116 BAX mutant cell lines and we detected BAK expression in HCT116 cell lines (**Fig. 1g, Suppl. Fig. 2b**). BAK expression is detected at similar levels between the HCT116 BAX mutant cell lines. We also added a new experiment comparing BAK KO with wild type CALU-6 cells and we found BDM19 potentiated apoptosis with ABT-263 in BAK KO CALU-6 cells similarly to wild type CALU-6 cells but to a lesser degree (**Fig. 7i**). This result suggests that BAK levels can contribute to apoptosis induced by BDM19, supposedly through interactions with activated BAX and formation of BAX-BAK oligomers on the mitochondrial outer membrane.

8. The model in Figure S2c is not as transparent as Figure 7j. At first glance, figure is not intuitive; why draw “minor apoptosis” since the argument is that the cells are resistant to death? Step-wise BAX on left vs. right sides of Fig. S2c was not explained in the text or legend. Dissociation of inactive dimer is not directly depicted in either model; is their doubt that dissociation is needed prior to death?

We thank the reviewer for this suggestion. We have improved the model; depicted resistance to death for the right side, included dissociation of BAX dimer and explained better the model in the legend. This model now is in supplementary figure 3c.

9. In Figure 7a and 7e legend, indicate how “cell viability” was measured (celltiter glo, caspase3/7 activity).

We indicated cell-titer glow in the legend for Fig 7a,e

10. Figure 4 legend is incorrectly labeled as Figure 3.

We corrected it.

11. Figure key and significance of data presented in Fig. S5a-d or legend are not articulated.

We have provided figure key and significance figure Fig. S5a-d legend which is now Fig. S8

12. Grammar error line 319.

We corrected “did not reduce”.

Figure with cellular thermal shift assay potentially could be incorporated into the final version of the manuscript after resolving dispute with Pelago Bioscience.

Figure. b-e Cellular thermal shift assay of BAX in MEF BAX BAK DKO reconstituted with **(b)** BAX WT or BAX mutants **(c)** K21E, **(d)** R134E and **(e)** R145E. Cells were treated with vehicle (DMSO) or 60 μ M of BDM19 for 1hr at room temperature (RT). Blots (top) are representative of three independent experiments. Melting curves for BAX (bottom) were generated by densitometric analysis. **(b-e)** Data are mean \pm SEM from three independent experiments.

Method of cellular thermal shift assay

DKO MEFs expressing BAX WT or mutants K21E, R134E and R145E cells were seeded in a 10 cm dish until approximately 85% confluent. The cells were washed and harvested in PBS. 7×10^6 Cells were incubated in 60 μ M of BDM19 dissolved in PBS for 1 hour at room temperature on a rotator. 50 μ l of the sample was transferred to PCR-tubes and heated in a Biorad C1000 Touch Thermal Cycler for 3 minutes using a temperature gradient (50, 52.1, 55.4, 59.4, 64.9, 69.2, 72.1, and 74°C). Cells remaining at room temperature (25°C) served as a control. Cells were lysed by three cycles of freeze thaw using liquid nitrogen and centrifuged at 2×10^4 g for 15 minutes. The supernatants were collected, resolved by SDS-PAGE on 4-12% NuPage gels in a 1X LDS/DTT loading buffer (Life Technologies) and immunoblotted using an N-terminal BAX antibody (Cell Signaling Cat. 2772). Proteins were visualized using the Odyssey Infrared Imaging System (LI-COR Biosciences), fluorescence intensity quantified using the Image Studio 3.1 software, and normalized to 25°C (100%) and 74°C (0%).

REVIEWERS' COMMENTS

Reviewer #1 (Remarks to the Author):

The authors made a lot of efforts to identify the structural basis of Bax dimer activation through mutation and SAR. Now it is very credible. I saw the new MD parameters. RMSD is always the most representative and convincing parameter in MD. Its unbalance is not ideal for the following analysis. Anyway, the completed wet experiments are sufficient enough. I recommend acceptance.

Reviewer #2 (Remarks to the Author):

I liked the MS a lot already before revision and now even more. The authors have addressed my points about non-cancer cells, the role of Venetoclax, the concentration of Bax used and the role of the various Bax mutations in AML patients and their role in Bax function and interplay with drug molecules and disturbances of the apoptotic pathways.

I think they authors have addressed that all very well and improved the MS substantially, especially also with respect to all work and additional information they have gathered to address the various severe points addressed by my four colleagues. I think they managed to address those points very well by carrying out work intensive additional work and improving and expanding the MS accordingly.

I think it is a really nice piece of research with high importance in the field of mitochondrial apoptosis and related cancer and drugs with focus on an unusual target, namely Bax and not the main one, Bcl-2.

Reviewer #3 (Remarks to the Author):

The clarifications and additional experiments the authors provided have answered my concerns and questions. Now the revised paper is much strengthened and ready to be published in Nature Communications in my opinion.

Reviewer #4 (Remarks to the Author):

My questions and concerns were addressed. I have no further comments.

We would like to thank the reviewers for their careful review of our work, complementary remarks and recommending acceptance. There are no more comments that need a response.

REVIEWERS' COMMENTS

Reviewer #1 (Remarks to the Author)

The authors made a lot of efforts to identify the structural basis of Bax dimer activation through mutation and SAR. Now it is very credible. I saw the new MD parameters. RMSD is always the most representative and convincing parameter in MD. Its unbalance is not ideal for the following analysis. Anyway, the completed wet experiments are sufficient enough. I recommend acceptance.

Reviewer #2 (Remarks to the Author):

I liked the MS a lot already before revision and now even more. The authors have addressed my points about non-cancer cells, the role of Venetoclax, the concentration of Bax used and the role of the various Bax mutations in AML patients and their role in Bax function and interplay with drug molecules and disturbances of the apoptotic pathways.

I think they authors have addressed that all very well and improved the MS substantially, especially also with respect to all work and additional information they have gathered to address the various severe points addressed by my four colleagues. I think they managed to address those points very well by carrying out work intensive additional work and improving and expanding the MS accordingly.

I think it is a really nice piece of research with high importance in the field of mitochondrial apoptosis and related cancer and drugs with focus on an unusual target, namely Bax and not the main one, Bcl-2.

Reviewer #3 (Remarks to the Author):

The clarifications and additional experiments the authors provided have answered my concerns and questions. Now the revised paper is much strengthened and ready to be published in Nature Communications in my opinion.

Reviewer #4 (Remarks to the Author):

My questions and concerns were addressed. I have no further comments.